# UniAG: Unified Anomaly Generation via Local Spatial-Texture Alignment Diffusion Model

## Abstract

Few-shot Anomaly Generation (FSAG) aims to enhance anomaly detection by generating realistic and diverse anomalies from a limited set of anomalous examples, addressing the challenge of scarce anomalous data in real-world scenarios. However, existing FSAG methods require training separate models for different anomaly types, leading to low training and deployment efficiency. Most importantly, the lack of sufficient realism and diversity limits the performance of anomaly detectors trained on them. To overcome these limitations, we propose UniAG, a unified model capable of generating realistic and diverse anomalies across multiple categories, thereby improving both generation efficiency and anomaly detection performance. Specifically, we propose a deep copy–paste anomaly generation strategy in which a Spatial-Texture Alignment Diffusion model (STA-DM) learns to fill local region masks with anomaly textures corresponding to user-specified categories. We further propose a novel generation condition with explicit spatial–category guidance instead of text embeddings for diffusion models, enabling realistic and diverse generation. Experimental results show that UniAG outperforms existing methods both in anomaly generation quality and in downstream anomaly detection performance. Notably, we achieve a new state-of-the-art anomaly localization AUROC / AP performance **99.2** / **81.0** with only **4** anomaly examples and **500** generated samples for each anomaly on the comprehensive MVTec AD dataset.

## 1 Introduction

Recently, anomaly detection has gained significant development and plays a vital role in industrial manufacturing, medical pathology analysis, and video surveillance Liu et al. (2024); Yang et al. (2021; 2022). However, in real-world application scenarios, the scarcity of anomaly samples and the diversity of anomaly patterns pose a major challenge to anomaly detection. Existing unsupervised anomaly detection methods have limited performance in anomaly localization Jin et al. (2025) and can not deal with the task of anomaly classification Hu et al. (2024). To address these issues, few-shot anomaly generation (FSAG) has gained increasing attention and can be divided into three categories: 1) Copy-paste-based methods Zavrtanik et al. (2021a); Li et al. (2021); Zhang et al. (2023a) crop and paste patches from existing normal or anomaly region onto normal samples, which exhibit significant semantic gaps or limited anomaly modes in the generated samples; 2) GAN-based methods Niu et al. (2020); Duan et al. (2023); Zhang et al. (2021); Liu et al. (2023a) are efficient in good perceptual quality, but difficult to capture the real anomaly distribution; 3) Text-to-image generation methods Hu et al. (2024); Gui et al. (2024); Jin et al. (2025) are recently emerging state-of-the-art diffusion-based FSAG methods, which mainly learn a textual embedding for generating each type of anomaly. Existing FSAG methods have demonstrated better performance than unsupervised approaches in anomaly localization and classification Hu et al. (2024); Jin et al. (2025). However, they suffer from the following limitations that call for refinement. First, they require training a separate model for each anomaly category, leading to inefficiencies in both training and generation. Second, generating anomalies using the full anomalous foreground and normal background can result in unstable outputs, including the disappearance of small-object anomalies and distortion of the background. Ultimately, due to the ambiguity of textual descriptions for rare, specialized, and long-tailed anomaly patterns, current state-of-the-art text-to-image diffusion models struggle to generate anomalies that are highly diverse and realistic. The dissimilarity from real anomalies introduces a

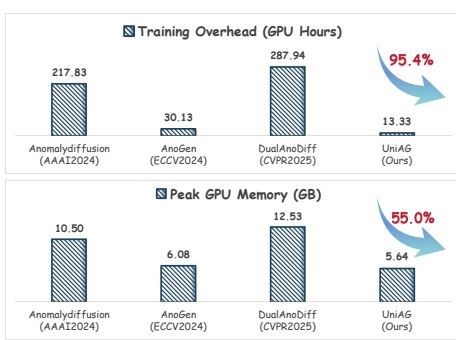 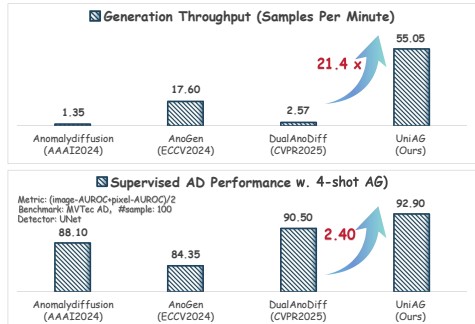

Figure 1: UniAG generates multi-class anomalies with a unified model, which improves training efficiency while facilitating model deployment and user-specified generation. It also improves generation throughput and reduces GPU memory usage during inference. By leveraging a novel locally focused generation condition with explicit spatial–category guidance, UniAG generates more realistic and diverse anomaly patterns, achieving superior anomaly detection performance compared to state-of-the-art single-class FSAG models under equivalent data volume and detection settings.

domain gap between training and testing, leading to degraded detection performance. To this end, we propose UniAG, a *deployment-friendly*, *efficient*, and *effective* FSAG approach. As illustrated in Fig. 1, UniAG enables the generation of multiple anomaly categories using a single unified model, reducing the training time to $1/20$ of the original and achieving over a $20\times$ speedup in generation with minimal inference-time GPU memory overhead compared to the latest method Jin et al. (2025). Importantly, by improving the realism and diversity of generated samples, UniAG delivers superior performance in both anomaly generation and downstream detection tasks. To this end, we propose a simple yet effective deep copy–paste strategy that synthesizes specified anomaly textures within designated regions and seamlessly blends them into randomly selected normal images to generate anomaly samples. To learn the distribution of real anomalous textures from scarce data, we introduce the Spatial-Texture Alignment Diffusion Model (STA-DM), which incorporates spatial and category cues as texture-related generation conditions, replacing the conventional textual embeddings. STA-DM extracts anomaly embeddings from input to ensure category-consistent generation in multi-class settings and incorporates trainable control layers that allow explicit spatial and semantic guidance from extracted spatial-category embeddings for realistic and diverse anomaly generation. Experiments on the widely used MVTec AD dataset show UniAG outperforms existing FSAG methods in both generation quality and subsequent anomaly detection performance.

In brief, our contributions can be summarized as follows:

- We analyze the limitations of existing FSAG methods and introduce a simple yet effective *deep copy-paste* generation strategy to address them. Further, we propose UniAG, a local anomalous texture-enhancing FSAG framework. *To the best of our knowledge, UniAG is the first to support unified and controllable anomaly generation that faithfully simulates real anomalous samples while significantly improving training and generation efficiency.*

- We propose a novel *Spatial-Texture Alignment Diffusion Model (STA-DM)*, which incorporates essential spatial and category information as anomaly generation conditions—in place of the commonly used textual conditions—to enhance the realism and diversity of generated anomalies for improved downstream detection performance.

- Extensive experiments on MVTec AD and cross-domain composite dataset (VisA, Brain-MRI, HeadCT) demonstrate that UniAG achieves superior generation quality and delivers consistent gains in downstream anomaly detection, localization, and classification tasks.

## 2 RELATED WORK

### 2.1 ANOMALY GENERATION

FSAG methods generate new anomaly samples from few-shot anomaly samples, enhancing supervised anomaly detection performance. We classify existing FSAG methods into **3** categories: (1) Copy-paste-based methods, (2) GAN-based methods, and (3)Text-to-image generation methods.

Figure 2: Conceptual illustration of the advantages of UniAG (bottom) over existing methods (top). Unlike copy–paste approaches (a, b), our proposed UniAG explicitly learns anomaly texture distributions, producing realistic and diverse samples. By simulating anomalous texture in few-shot real references, it reduces the gap between training and testing data and avoids the instability caused by overreliance on textual semantic similarity (c). In contrast to holistic generation methods, UniAG adopts a localized generation–blending strategy, which enables faithful synthesis of small-area anomalies (d) while maintaining intact normal object and background regions (e).

**Copy-paste based Methods.** DRAEM Zavrtanik et al. (2021a), Cut-Paste Li et al. (2021), PRN Zhang et al. (2023a) crop and paste local irrelevant normal or real anomalous textures into normal samples as anomaly samples in an efficient way.

**GAN-based Methods.** SDGAN Niu et al. (2020) and Defect-GAN Zhang et al. (2021) generate anomalies on normal samples by learning the anomaly distribution. However, they require a large amount of anomaly data and cannot generate anomaly masks. DFMGAN Duan et al. (2023) transfers a pre-trained StyleGANv2 Karras et al. (2020) on normal samples to the anomaly domain and then trains a specific generation model for each anomaly category on limited anomaly samples.

**Text-to-image Generation Methods.** AnomalyDiffusion Hu et al. (2024) and AnoGen Gui et al. (2024) leverage textual inversion Gal et al. (2023) to learn text embeddings from few-shot anomalies for generation. DualAnoDiff Jin et al. (2025) further introduces a dual diffusion framework Rombach et al. (2022) to jointly generate overall anomalous images and their corresponding anomaly regions. SeaS Dai et al. (2024) uses an unbalanced abnormal prompt to disentangle anomaly attributes and assign them to specific tokens, enabling the generation of diverse anomaly types along with normal products. Similar to Hu et al. (2024), it trains a separate mask-generation model.

**Limitation of Current FSAG Methods.** As shown in Fig. 2, we summarize the limitations of existing methods as follows: (1) Training a separate model for each anomaly type is inefficient, especially when the number of categories is large; (2) It is difficult to balance realism and diversity, *e.g.*, anomalies generated by simple copy–paste strategy either lack realism or diversity, leading the trained detection model to suffer from underfitting or overfitting; (3) Textual embedding conditions in generation result in ambiguous anomaly semantics, and together with the neglect of spatial information, they reduce the realism and validity of generated samples; and (4) Simultaneously generating anomalous foregrounds and normal backgrounds as a whole often yields unreliable results for small-area anomalies (weakened or even disappear) and degrades background realism.

## 2.2 ANOMALY DETECTION

Unsupervised anomaly detection methods assume that only normal data are accessible, which can be broadly divided into three categories. Reconstruction-based methods Gong et al. (2019); Ristea et al. (2022); Zavrtanik et al. (2021b); Zhang et al. (2024); He et al. (2024) aim to reconstruct normal data and detect anomalies based on reconstruction errors. Synthesis-based methods Li et al. (2021); Zavrtanik et al. (2021a); Chen et al. (2024a;b) generate anomalous samples to augment training and improve detection. Embedding-based methods Bergmann et al. (2020); Defard et al. (2021); Deng & Li (2022); Tien et al. (2023); Roth et al. (2022); Liu et al. (2023b) leverage feature representations to distinguish normal and abnormal patterns. However, these methods face several challenges, including insufficient pixel-level anomaly localization performance Zhou et al. (2024), inability to support fine-grained anomaly classification Jin et al. (2025), and lower inference efficiency compared with simple supervised networks (such as Unet) Zhang et al. (2023a); Yao et al. (2023).

Figure 3: Left: UniAG employs a deep copy–paste strategy to learn spatial–category–correlated local anomaly textures during training. At inference, it generates realistic and diverse anomalous patches conditioned on the specified category and spatial region, which are subsequently blended into normal images. Right: The generated anomalies faithfully capture the learned texture distribution, yielding realistic and diverse samples even from the same input.

## 3 METHOD

We formalize multi-class FSAG setting as follows. Denoting $\mathcal{A}_n$ as $k$-shot anomalous samples with paired image $s_n^*$ and mask $m_n^*$ for the $n$-th anomaly category:

$$\mathcal{A}_n = \{(s_n^1, m_n^1), \cdots, (s_n^k, m_n^k)\},$$

our goal is to learn a single model $\mathcal{M}_\theta$ with parameter $\theta$ that generate pseudo anomaly dataset $D_n \subset \mathbb{D}$ corresponding to the $n$-th anomaly category. The whole generation process can be expressed as:

$$\mathbb{D} = \bigcup \mathcal{M}_\theta(\mathcal{A}_n), \tag{1}$$

The *realism* and *diversity* of $\mathbb{D}$ are crucial for the effective supervised training of anomaly detection models on it and improved final performance Duan et al. (2023); Hu et al. (2024); Jin et al. (2025).

### 3.1 DEEP COPY-PASTE STRATEGY

To address the above limitations, we propose a simple yet effective one-solution-for-all generation strategy that "copies" pseudo anomaly patches in a *learnable* way and blends them into normal images as new anomalous samples. Our proposed deep copy-paste generation strategy offers **4** key advantages: (1) Unlike Zavrtanik et al. (2021a); Li et al. (2021) introduce anomaly noise or heterogeneous patches as anomaly textures, or Zhang et al. (2023a) repeats a limited set of anomaly patches, we propose an anomaly-driven generation strategy to learn texture distribution, ensuring *realistic* and textit{diverse} generated anomalies; (2) A new generation condition that integrates anomaly categories and spatial information is proposed to replace textual embeddings to enhance generation capability. (3) In contrast to full-image inpainting approaches Hu et al. (2024); Gui et al. (2024) that may lead to inferior results of small anomalies, we propose a local-focused anomaly inpainting strategy that enables reliable generation of small anomalies while mitigating the impact on the normal background Jin et al. (2025). The overall deep copy-paste process is illustrated in Fig. 3.

**Deep Anomaly Texture Copying.** We train a spatial-texture alignment diffusion model $\mathcal{M}_\theta^{sta}$ to learn anomaly textures corresponding to specified anomaly category and region mask. To enhance the generation of small anomalies, we extract foreground anomaly regions through mask's bounding box cropping and apply random affine transformations to enhance data diversity. Denoting the resized $\hat{s_n^i}, \hat{m_n^i}$ as processed pairs, $id(\cdot)$ as the anomaly identifier, this process can be expressed as:

$$\hat{s_n^i}, \hat{m_n^i} = Resize(Augment(Crop(s_n^i, m_n^i)), 256), \tag{2}$$

$$\theta^* = \min \sum_{n=1}^{|\mathcal{A}|} \sum_{i=1}^{k} \mathcal{L}_{gen}(\hat{s_n^i}, \mathcal{M}_\theta^{sta}(id(\mathcal{A}_n), \hat{m_n^i})). \tag{3}$$

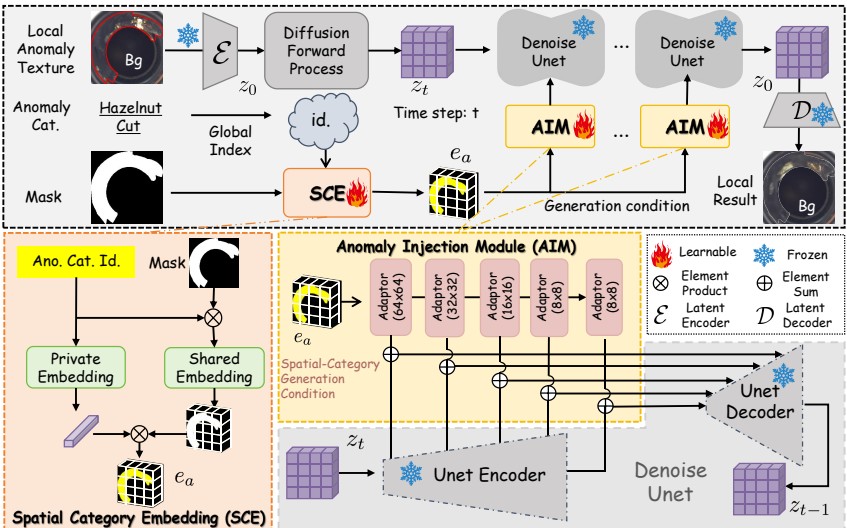

Figure 4: Illustration of spatial-texture alignment diffusion model (STA-DM). We adopt a local-cropping-based texture-focused inpainting approach and replace the conventional textual condition with a novel anomaly embedding $e_a$, which incorporates both spatial and categorical information. Spatial category embedding (SCE) and Anomaly Injection Adapter (AIA) are proposed to extract $e_a$ and inject it into the generation process of the diffusion model, respectively.

We use the global category index of the anomaly as $id(\mathcal{A}_n)$ (*i.e.*, $n$), and $\mathcal{L}_{gen}$ is the training loss of $\mathcal{M}_\theta^{sta}$ to generate the target $\hat{s_n^i}$ corresponding to $\hat{m_n^i}$ and $id$.

**Spatial–Category Guided Pasting.** During inference, local anomaly patches are sampled from noise in the context of the background within the crop-box, which can be formally expressed as:

$$z_t' = z_a^t \cdot (1 - \hat{m}) + z_t \cdot \hat{m}, \tag{4}$$

$$z_a^t = \textit{Diffuse}(\hat{s}, t). \tag{5}$$

Here, $(\hat{s}, \hat{m})$ denotes the cropped and augmented local anomaly sample, $z_t$ is the predicted noise at timestep $t$, and $z_a^t$ denotes noise of $\hat{s}$ in the latent diffusion forward process *Diffuse* at timestep $t$. We combine the predicted anomaly noise with the diffused background as the input $z_t'$ for the subsequent denoising step. To ensure stable anomaly generation, particularly for small regions, our approach differs from existing full-image inpainting methods Hu et al. (2024); Gui et al. (2024) in two key aspects. First, we adopt local inpainting to minimize background interference on small anomalous foregrounds. Second, instead of using diffused normal images as background, we initialize with diffused local anomaly patches (*i.e.*, $z_a^t$), alleviating inconsistencies caused by domain misalignment between training and testing Lugmayr et al. (2022). Following Zhang et al. (2023a), the generated anomaly patches are randomly placed onto foreground regions of normal samples. Although UniAG already produces diverse and realistic anomalies with the same mask from few-shot samples (Fig. 3, right), we further enhance diversity through spatial and appearance augmentations. Details of the random transformations, parameter settings, and comparisons with existing methods on masks derivation are provided in Appendix E.3. We also evaluate several blending strategies, including Poisson blending, fusion autoencoder, normal diffusion, and direct pasting, and find that the simplest way performs best (Please refer to anomaly fusion ablation for more details).

## 3.2 SPATIAL-TEXTURE ALIGNMENT DIFFUSION MODEL

We design a novel anomalous patch generation model as shown in Fig. 4, which consists mainly of the Spatial Category Embedding (SCE) and Anomaly Injection Adapter (AIA) to realize *spatial-aware multi-category realistic* and *diverse* anomalous patch generation. We detail them as follows.

**Spatial-Category Embedding.** Existing anomaly-driven FSAG methods Duan et al. (2023); Hu et al. (2024); Gui et al. (2024) are single-category-based and require training a specific weight for each type of anomaly, which are not friendly for training and generation in practical applications.

Table 1: ($IS \uparrow$, $IC$-$L \uparrow$, $FID \downarrow$) comparison of generated samples. **Bold** represents best results.

| Subset | Anodiff(AAAI'24) | | | AnoGen(ECCV'2024) | | | DualAnoDiff(CVPR'25) | | | SeaS(ICCV'25) | | | UniAG(ours) | | |
|---|---|---|---|---|---|---|---|---|---|---|---|---|---|---|---|
| | IS↑ | IC-L↑ | FID↓ | IS↑ | IC-L↑ | FID↓ | IS↑ | IC-L↑ | FID↓ | IS↑ | IC-L↑ | FID↓ | IS↑ | IC-L↑ | FID↓ |
| cable | 2.40 | 0.57 | 0.67 | 2.60 | 0.54 | 0.59 | 2.01 | 0.27 | 3.06 | 2.32 | 0.57 | 0.75 | **2.68** | **0.58** | **0.31** |
| carpet | 2.03 | 0.54 | 5.68 | **2.33** | 0.53 | 5.90 | 2.06 | **0.58** | 6.44 | 2.01 | 0.50 | 5.85 | 2.29 | 0.57 | **1.08** |
| toothbrush | 2.80 | 0.57 | 0.26 | 2.26 | **0.53** | 0.28 | 3.21 | 0.58 | 3.28 | 2.77 | 0.59 | 1.62 | **3.31** | **0.62** | **0.06** |
| transistor | **2.84** | 0.51 | 0.35 | 2.54 | 0.52 | 0.68 | 2.46 | 0.47 | 1.59 | 2.79 | 0.53 | 1.70 | **2.84** | 0.52 | **0.09** |
| bottle | **2.46** | 0.42 | 0.28 | 2.48 | 0.51 | 0.24 | 2.42 | **0.47** | 1.80 | 2.11 | **0.47** | 0.32 | 2.30 | 0.46 | **0.12** |
| tile | 2.05 | 0.56 | 1.28 | 2.17 | 0.52 | 1.02 | 2.35 | **0.61** | 13.3 | **2.76** | 0.57 | 1.33 | 2.19 | 0.53 | **0.39** |
| zipper | 1.83 | 0.47 | 0.36 | 2.18 | 0.49 | 0.33 | **2.23** | 0.51 | 9.01 | 2.06 | 0.49 | 0.69 | 2.09 | **0.54** | **0.31** |
| wood | 1.74 | 0.40 | 1.31 | 2.62 | 0.44 | 1.24 | 2.14 | 0.47 | 2.88 | 2.30 | 0.46 | 8.45 | **2.68** | **0.49** | **0.57** |
| hazelnut | 2.80 | 0.49 | 0.35 | 2.43 | 0.49 | 0.38 | 2.53 | 0.53 | 2.09 | **2.93** | **0.56** | 0.81 | 2.74 | 0.53 | **0.10** |
| grid | **3.03** | **0.56** | 8.22 | 2.83 | 0.51 | 4.37 | 2.83 | 0.51 | 5.85 | 2.46 | 0.51 | 3.94 | 3.01 | 0.53 | **2.57** |
| screw | 1.86 | 0.31 | 0.38 | 2.55 | 0.41 | 0.36 | 2.53 | 0.45 | 2.22 | **2.64** | **0.52** | **0.09** | 2.60 | 0.49 | 0.19 |
| capsule | 1.93 | 0.30 | 0.39 | 2.66 | **0.47** | 0.34 | 2.61 | 0.41 | 1.42 | 2.45 | **0.47** | 0.29 | **2.71** | 0.42 | **0.19** |
| leather | 2.04 | 0.42 | 6.82 | 2.28 | 0.41 | 7.19 | 2.66 | **0.52** | 2.71 | **2.71** | 0.51 | 8.20 | 2.32 | 0.43 | **1.49** |
| metal_nut | 2.38 | 0.52 | 0.53 | 2.33 | 0.50 | 1.09 | 2.23 | **0.55** | 1.69 | 1.71 | 0.49 | 1.39 | **2.44** | 0.51 | **0.21** |
| pill | 2.46 | 0.42 | 0.50 | 2.35 | 0.46 | 0.70 | **2.53** | 0.50 | 7.22 | 2.10 | **0.57** | 1.05 | 2.50 | 0.45 | **0.30** |
| **Mean** | 2.31 | 0.47 | 1.83 | 2.44 | 0.49 | 1.65 | 2.45 | 0.49 | 5.04 | 2.41 | **0.52** | 2.43 | **2.58** | 0.51 | **0.53** |

To address this, we consider the anomaly category along with the region mask as the FSAG model's input to generate corresponding local anomaly textures. Inspired by Vaswani et al. (2017); Akata et al. (2015), we design a novel Spatial Category Embedding (SCE) module to integrate spatial region and category information to construct anomaly embedding with explicit spatial–category guidance, which serves as the generation condition for the diffusion model. SCE consists of two parts: category-shared embedding $e_s$ and category-private embedding $e_p$. We replace the foreground (value = 1) region in the corresponding mask with the anomaly identifier $id(\mathcal{A}_n)$, and obtain the category-shared embedding containing the anomaly spatial and category information through shared embedding blocks $ZC$ consisting of $1 \times 1$ convolution layer with both weight and bias initialized to zeros, which can be formalized as:

$$e_s = ZC(\hat{m_n} \cdot n). \tag{6}$$

In addition, we set a private learnable embedding $e_p \in \mathbb{R}^{320}$ for each category to enhance the model's ability to distinguish different anomalies. And combine it with the category-shared embedding with channel-wise element product to obtain the spatial-category generation condition:

$$e_a = e_s \cdot e_p. \tag{7}$$

Compared to ambiguous textual conditions, $e_a$ provides precise spatial cues that guide the model to focus on generating realistic anomaly texture. Meanwhile, $e_a$ encodes the anomaly category to support multi-class anomaly generation and outputs consistent textures to given anomaly categories.

**Anomaly Injection Adapter.** Inspired by structure-conditioned diffusion methods Zhang et al. (2023b); Li et al. (2024), we adopt a trainable copy of the denoise UNet encoder in the latent diffusion model Rombach et al. (2022) to learn the spatial-texture alignment of anomalous patterns. For each layer in the denoise UNet encoder, zero convolutions $ZC(\cdot; \theta)$ are used to connect the output of the original frozen branch $\mathcal{F}(\boldsymbol{x}; \theta)$ and the trainable branch $\mathcal{T}(\boldsymbol{x}; \theta_t)$. Denoting the output of the AIA block as $y$, the forward process can be expressed as:

$$\boldsymbol{y} = \mathcal{F}(\boldsymbol{x}; \theta) + ZC(\mathcal{T}(\boldsymbol{x} + e_a; \theta_t); \theta). \tag{8}$$

We discard textual embedding (simply set text as "") and use $e_a$ that contains both spatial and category anomaly information as a condition to generate the corresponding textures. Based on the original architecture Zhang et al. (2023b), we modify the prompt input and training strategy for the anomaly generation task. The optimization objectives of training are as follows:

$$\mathcal{L}_{gen} = \mathbb{E}_{\boldsymbol{z}_0, \boldsymbol{t}, \boldsymbol{e_a}, \epsilon \sim \mathcal{N}(0,1)} \left[ \| \epsilon - \epsilon_\theta \left( \boldsymbol{z}_t, \boldsymbol{t}, \boldsymbol{e_a} \right) \|_2^2 * \hat{m} \right]. \tag{9}$$

AIA integrates the extracted $e_a$ into the diffusion model during training and introduces fine-tuning parameters to adapt the model accordingly. Notably, the spatial and appearance augmentations employed during training effectively decouple anomaly patterns into spatial and texture components. Guided by the spatial cues in $e_a$, the diffusion model is encouraged to explore novel texture patterns that resemble real anomalies, thereby improving the diversity of generated samples.

## 4 EXPERIMENTS

**Dataset and baseline.** Following the latest representative FSAG methods Duan et al. (2023); Hu et al. (2024); Gui et al. (2024); Jin et al. (2025), we mainly conduct experiments on the widely used

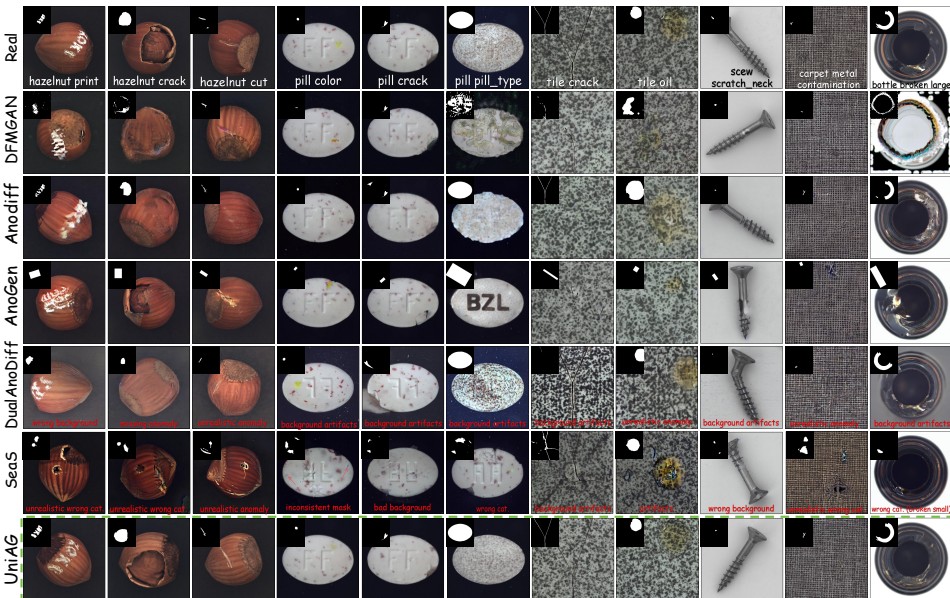

Figure 5: Visual comparison with generated samples on 11 anomaly categories. Please zoom in for a better view of anomaly realism and category, mask alignment, and background consistency.

Table 2: Quantitative result for image and pixel level anomaly detection on 4-shot MVTec AD dataset by training a simple UNet on the **100** generated data from different FSAG methods. The best image and pixel-level performance is highlighted in **bold**.

| Category | Anodiff(AAAI'24) $AUC_i$ | $AP_i$ | $AUC_p$ | $AP_p$ | AnoGen(ECCV'2024) $AUC_i$ | $AP_i$ | $AUC_p$ | $AP_p$ | DualAnoDiff(CVPR'25) $AUC_i$ | $AP_i$ | $AUC_p$ | $AP_p$ | SeaS(ICCV'25) $AUC_i$ | $AP_i$ | $AUC_p$ | $AP_p$ | UniAG(Ours) $AUC_i$ | $AP_i$ | $AUC_p$ | $AP_p$ |
|---|---|---|---|---|---|---|---|---|---|---|---|---|---|---|---|---|---|---|---|---|
| bottle | 97.7 | 99.3 | 94.6 | **81.6** | 96.6 | 98.8 | 75.6 | 57.4 | 98.8 | **99.6** | **95.0** | 76.5 | 96.1 | 98.7 | 72.1 | 50.4 | 97.5 | 99.1 | 94.1 | 79.7 |
| cable | **99.5** | **99.5** | 93.2 | **74.2** | 95.8 | 97.0 | 81.2 | 57.5 | 91.3 | 93.8 | 87.8 | 64.6 | 93.4 | 95.6 | 76.4 | 60.5 | 96.8 | 97.4 | 89.2 | 62.2 |
| capsule | 77.4 | 94.3 | 74.8 | 12.6 | 78.9 | 94.0 | 62.9 | 19.4 | 88.8 | 92.3 | 63.4 | 13.7 | 92.8 | 98.2 | 59.7 | 28.6 | **96.1** | **99.0** | **77.8** | **32.2** |
| carpet | 64.8 | 79.4 | 79.8 | 17.8 | 90.0 | 96.6 | 62.7 | 48.2 | **95.6** | **98.4** | **97.5** | **80.2** | 91.2 | 97.2 | 87.3 | 64.2 | 84.8 | 93.6 | 85.9 | 59.3 |
| grid | 97.4 | 98.9 | 89.8 | 32.2 | 96.0 | 98.0 | 71.8 | 29.5 | 93.3 | 96.8 | 89.2 | 38.0 | 99.5 | 99.7 | 81.0 | 40.8 | **99.9** | **100.** | **87.8** | **52.2** |
| hazelnut | 95.9 | 97.5 | 88.5 | 70.0 | 98.0 | 98.8 | 92.7 | 61.6 | **100.** | **100.** | 98.1 | 91.5 | 99.7 | 99.8 | 82.1 | 64.2 | **100.** | **100.** | **98.2** | **93.9** |
| leather | 99.1 | 99.7 | 91.9 | 65.9 | **100.** | **100.** | 82.4 | 60.1 | 99.0 | 99.7 | 86.3 | 68.6 | 99.2 | 99.7 | 82.8 | 55.8 | **100.** | **100.** | **97.8** | **72.9** |
| metal_nut | 99.8 | 99.9 | 98.7 | 96.4 | 99.5 | 99.9 | 98.3 | 89.1 | 98.6 | 99.7 | 96.7 | 88.2 | 99.4 | 99.8 | 98.2 | 95.3 | **100.** | **100.** | **99.6** | **98.3** |
| pill | 93.2 | 98.6 | 94.9 | 79.3 | 97.5 | 99.4 | 96.4 | 78.9 | 92.4 | 98.3 | 95.5 | 71.6 | 90.5 | 98.0 | 67.9 | 52.2 | **99.2** | **99.9** | **98.7** | **84.0** |
| screw | 12.5 | 53.7 | 56.8 | 1.71 | 37.3 | 69.1 | **63.7** | 7.70 | 58.3 | 82.8 | 87.4 | 25.4 | **78.9** | **90.7** | 55.3 | **32.3** | 62.5 | 85.9 | 52.0 | 22.6 |
| tile | **100.** | **100.** | 96.6 | 91.0 | **100.** | **100.** | 90.9 | 72.0 | **100.** | **100.** | 95.2 | 82.5 | 99.6 | 99.8 | **98.9** | 90.6 | **100.** | **100.** | 97.6 | 94.5 |
| toothbrush | 95.8 | 98.2 | 73.2 | 27.9 | 93.6 | 97.0 | 60.6 | 22.5 | 96.5 | 98.5 | 71.8 | 25.8 | 92.3 | 97.1 | 80.8 | 43.0 | **100.** | **100.** | **91.4** | **65.6** |
| transistor | **99.9** | **99.8** | **94.3** | **81.2** | 98.0 | 95.8 | 66.0 | 38.6 | 88.4 | 88.5 | 81.8 | 47.1 | 93.9 | 93.1 | 75.1 | 34.8 | 99.1 | 97.8 | 93.7 | 78.0 |
| wood | 98.6 | 99.4 | 94.6 | 75.7 | **99.5** | **99.8** | 86.9 | 68.1 | **99.5** | **99.8** | 83.5 | 60.8 | 96.6 | 98.7 | 86.3 | 60.6 | 98.0 | 98.9 | **96.3** | **80.4** |
| zipper | **100.** | **100.** | 89.5 | 68.5 | 99.6 | 99.9 | 58.2 | 38.9 | **100.** | **100.** | 86.9 | 65.5 | 99.5 | 99.8 | 73.3 | 54.1 | **100.** | **100.** | **93.6** | **73.4** |
| **Mean** | 88.8 | 94.5 | 87.4 | 58.4 | 92.0 | 96.3 | 76.7 | 50.0 | 93.3 | 96.9 | 87.7 | 60.0 | 94.8 | 97.7 | 78.5 | 55.2 | **95.6** | **98.1** | **90.2** | **69.9** |

**MVTec AD** Bergmann et al. (2019) **dataset**. MVTec AD contains a substantially larger number of 15 objects and 73 anomaly categories than other datasets, making it a comprehensive benchmark for generation evaluation. We randomly select 4 samples from each anomaly category for FSAG training, and the rest for testing. We compare with state-of-the-art FSAG methods, including Anomalydiffusion (We refer to it as Anodiff for brevity.) Hu et al. (2024), AnoGen Gui et al. (2024), DualAnoDiff Jin et al. (2025), and SeaS Dai et al. (2024). To further validate the generalizability, we additionally report the generation and detection performance on a **cross-domain composite dataset** comprising **VisA** Zou et al. (2022) and two medical datasets (**HeadCT and BrainMRI** Salehi et al. (2021)). Besides, we compare with SOTA unsupervised methods Roth et al. (2022); Liu et al. (2023b); Zhang et al. (2024) on anomaly localization task as in Hu et al. (2024); Jin et al. (2025).

**Implementation details.** We use foreground masks provided in Zhang et al. (2023a) to obtain the object foreground position in the normal image. We set shape of input $(s_n, m_n)$ as $256 \times 256$ and the private learnable embedding $e_p \in \mathbb{R}^{320}$. More details can be found in Appendix.D.

**Metric.** Following previous work Duan et al. (2023); Hu et al. (2024); Jin et al. (2025), we use Inception Score (IS) and Intra-cluster pairwise LPIPS distance (IC-L) to evaluate the generation

Table 3: Anomaly localization (AUROC / AP) comparison with UNet and FSN (as in RealNet) on 500 samples against SOTA unsupervised methods using their official codes.

| Subset | Unsupervised Methods | | | | Supervised with FSAG Samples | | | |
|---|---|---|---|---|---|---|---|---|
| | DRAEM (ICCV2021) | Patchcore (CVPR2022) | SimpleNet (CVPR2023) | RealNet (CVPR2024) | DualAnoDiff (w. UNet) | SeaS (w. UNet) | **UniAG** (w. UNet) | **UniAG** (w. FSN) |
| Carpet | 95.5 / 53.5 | 99.1 / 66.0 | 98.2 / 41.9 | 98.9 / 58.7 | **99.3** / **86.9** | 86.3 / 64.4 | 97.8 / 73.0 | **99.3** / 79.6 |
| Grid | 99.7 / 65.7 | 98.7 / 24.6 | 98.8 / 33.9 | 99.5 / 55.5 | 97.6 / 34.3 | 94.1 / 60.4 | 97.7 / 65.9 | **99.6** / **70.5** |
| Leather | 98.6 / 75.3 | 99.3 / 44.4 | 99.2 / 43.3 | 99.6 / 65.7 | 98.6 / 81.1 | 73.4 / 48.1 | **100.** / **82.3** | 99.8 / 78.4 |
| Tile | 99.2 / 92.3 | 95.9 / 57.0 | 97.0 / 65.0 | 98.2 / 84.1 | 99.3 / 93.8 | 99.0 / 87.2 | **100.** / **97.9** | 99.2 / 95.0 |
| Wood | 96.4 / 77.7 | 95.1 / 55.3 | 94.5 / 46.7 | 97.7 / 74.0 | **98.2** / 76.1 | 91.1 / 65.8 | **98.2** / **79.4** | 98.0 / 78.6 |
| Bottle | 99.1 / 86.5 | 98.6 / 76.0 | 98.0 / 70.7 | 99.2 / 87.0 | 98.0 / 85.6 | 85.8 / 67.7 | 98.5 / 85.1 | **99.6** / **94.1** |
| Capsule | 94.3 / 49.4 | 98.8 / 47.6 | 97.6 / 41.8 | 99.2 / 58.5 | 97.2 / 39.6 | 53.7 / 26.1 | 96.0 / 41.7 | **99.4** / **67.7** |
| Pill | 97.6 / 48.5 | 97.6 / 76.9 | 98.9 / 79.6 | 98.9 / 82.2 | 99.0 / 90.1 | 76.8 / 46.2 | **99.6** / **91.7** | 99.5 / 91.4 |
| Transistor | 90.9 / 50.7 | 96.4 / 67.7 | 97.9 / 67.6 | 97.7 / 71.9 | 97.9 / 83.9 | 72.7 / 36.9 | 97.7 / 86.4 | **98.7** / **87.8** |
| Zipper | 98.8 / 81.5 | 98.9 / 64.1 | 98.8 / 65.3 | 98.9 / 68.7 | 99.0 / **87.2** | 68.6 / 46.0 | 98.8 / 85.0 | **99.3** / 80.8 |
| Cable | 94.7 / 52.4 | 98.5 / 65.9 | **98.6** / 66.5 | 97.2 / 52.0 | 91.0 / 68.2 | 84.8 / 66.3 | 96.6 / **76.3** | 98.2 / 73.9 |
| Hazelnut | 99.7 / 92.9 | 98.7 / 56.2 | 99.3 / 48.5 | 99.2 / 70.9 | 99.5 / 95.1 | 94.4 / 77.8 | **100.** / **96.2** | 99.8 / 93.4 |
| Metal_nut | 99.5 / 96.3 | 98.4 / 87.6 | 98.5 / 89.2 | 97.7 / 80.9 | 99.1 / 95.3 | 97.7 / 93.9 | **100.** / **98.5** | 99.6 / 97.4 |
| Screw | 97.6 / **58.2** | 99.4 / 35.2 | 97.6 / 34.5 | **99.4** / 55.4 | 94.4 / 32.8 | 58.3 / 28.6 | 93.1 / 43.9 | 98.9 / 52.0 |
| Toothbrush | 98.1 / 44.7 | 98.7 / 38.1 | 98.9 / 41.5 | 98.9 / 62.6 | 94.4 / 62.1 | 76.0 / 50.8 | 97.8 / 75.3 | **99.3** / **73.8** |
| **Mean** | 97.3 / 68.4 | 98.1 / 57.6 | 98.1 / 55.7 | 98.7 / 68.5 | 97.5 / 76.1 | 80.8 / 57.7 | 98.2 / 79.4 | **99.2** / **81.0** |

Table 4: Comparison of averaged anomaly classification accuracy with **100** generated samples.

| Methods | Anodiff | AnoGen | DualAnoDiff | SeaS | UniAG |
|---|---|---|---|---|---|
| ACC. | 69.5 | 62.7 | 74.3 | 43.9 | **78.5** |

quality within anomalous region. In addition, we report commonly used generation evaluation metric Fréchet Inception Distance (FID) Heusel et al. (2017) between the generated samples and real test samples within masked regions. We use image and pixel-level AUROC and AP to evaluate the anomaly detection performance. We use accuracy to evaluate anomaly classification performance.

## 4.1 ANOMALY GENERATION COMPARISON

We report (IS $\uparrow$, IC-L $\uparrow$, FID $\downarrow$) on **100** samples generated by different FSAG methods in Tab. 1. UniAG generates anomaly samples with the highest realism and diversity. Visual comparison of their typical generated samples is provided in Fig. 5. AnomalyDiffusion often produces anomaly-missing samples (*e.g.*, tile crack). AnoGen struggles with inconsistency between generated and real samples (*e.g.*, pill crack). DualAnoDiff and SeaS exhibit background inconsistency with real images in generated samples, which may negatively affect AD performance. Notably, the results of SeaS reveal an issue in which the generated anomalous samples often correspond to incorrect categories. In contrast, the proposed UniAG produces diverse and realistic anomalous regions, preserves the integrity of normal backgrounds, and remains consistent with the specified categories.

## 4.2 ANOMALY DETECTION COMPARISON

Following Hu et al. (2024); Jin et al. (2025), we use AUROC and AP to evaluate the AD performance of UNet Ronneberger et al. (2015) trained on **100** generated samples. We use the average validation performance over the last 10% of training epochs and report the average result of 3 runs. Results in Tab. 2 show that UniAG outperforms current SOTA FSAG methods by (**2.3**, **1.2**, **2.5**, **9.9**) on (AUROC$_{image}$, AP$_{image}$, AUROC$_{pixel}$, AP$_{pixel}$), respectively. We further compare with **500** samples against current SOTA unsupervised AD methods in line with their evaluation protocols and report the (AUROC$_{pixel}$, AP$_{pixel}$) in Tab.3, following Hu et al. (2024); Jin et al. (2025). UniAG with simple UNet achieves better results over both SimpleNet (**0.1**, **23.7**) and DualAnoDiff (**0.7**, **3.3**). With the same detector FSN Zhang et al. (2024), UniAG outperforms RealNet by (**0.5**, **12.5**) proving its effectiveness. DualAnoDiff cannot provide a corresponding normal counterpart for generated anomaly samples, making it incompatible for reconstruction-based AD models such as FSN.

## 4.3 ANOMALY CLASSIFICATION COMPARISON

Following Duan et al. (2023); Jin et al. (2025), we employ ResNet-18 to train an anomaly classification model on the generated dataset. We then evaluate the classification accuracy on our test dataset as shown in Tab. 4. Our model achieves an object-averaged accuracy improvement of **4.2** over others, indicating that the anomaly samples we generate are more category realistic and consistent.

Table 5: Anomaly generation comparison of 100 generated samples on 4-shot composite dataset.

| Methods | Anodiff(AAAI'24) | | | AnoGen(ECCV'2024) | | | DualAnoDiff(CVPR'25) | | | SeaS(ICCV'25) | | | UniAG(ours) | | |
|---|---|---|---|---|---|---|---|---|---|---|---|---|---|---|---|
| | IS↑ | IC-L↑ | FID↓ | IS↑ | IC-L↑ | FID↓ | IS↑ | IC-L↑ | FID↓ | IS↑ | IC-L↑ | FID↓ | IS↑ | IC-L↑ | FID↓ |
| Object-Averaged | 2.37 | 0.50 | 1.07 | 2.41 | 0.50 | 0.99 | 2.41 | 0.49 | 4.09 | 2.39 | 0.51 | 2.12 | **2.61** | **0.57** | **0.43** |

Table 6: Image and pixel level anomaly detection comparison on 4-shot composite dataset. The best image and pixel-level performance is highlighted in **bold**. (100 training samples per anomaly.)

| Methods | Anodiff(AAAI'24) | | | | AnoGen(ECCV'2024) | | | | DualAnoDiff(CVPR'25) | | | | SeaS(ICCV'25) | | | | UniAG(Ours) | | | |
|---|---|---|---|---|---|---|---|---|---|---|---|---|---|---|---|---|---|---|---|---|
| | $AUC_i$ | $AP_i$ | $AUC_p$ | $AP_p$ | $AUC_i$ | $AP_i$ | $AUC_p$ | $AP_p$ | $AUC_i$ | $AP_i$ | $AUC_p$ | $AP_p$ | $AUC_i$ | $AP_i$ | $AUC_p$ | $AP_p$ | $AUC_i$ | $AP_i$ | $AUC_p$ | $AP_p$ |
| Object-Averaged | 89.3 | 93.0 | 95.3 | 72.8 | 88.0 | 91.2 | 92.9 | 68.1 | 89.3 | 93.2 | 95.2 | 73.1 | 89.4 | 93.4 | 92.1 | 68.3 | **90.0** | **93.7** | **96.0** | **73.8** |

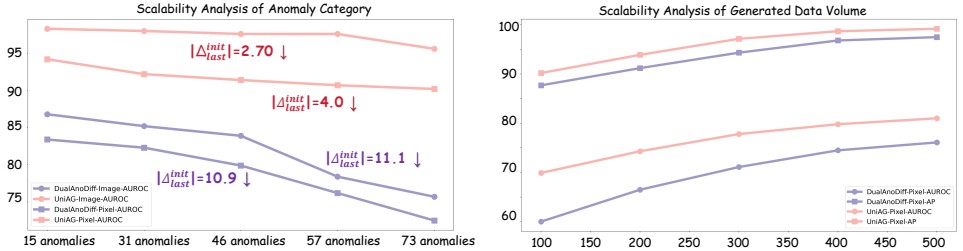

Figure 6: Scalability analysis. (Left) Category scalability: UniAG achieves superior multi-class generation with smaller performance drops as anomaly categories increase. (Right) Data scalability: UniAG improves steadily with larger data scales and outperforms DualAnoDiff consistently.

## 4.4 GENERATION SCALABILITY

We evaluate the *category scalability* of UniAG by first training on 3 objects (*bottle*, *carpet*, *leather*) with 15 anomaly categories, and then progressively expanding to the full MVTec AD dataset. The average performance on these 3 categories is shown in Fig. 6 (left). UniAG outperforms DualAnoDiff with smaller performance drops as categories increase. For *data-scale scalability*, we vary the amount of generated data (Fig. 6, right). UniAG sustains better performance over DualAnoDiff across various data scales, and larger generated datasets consistently improve detection accuracy.

## 4.5 CROSS-DOMAIN EVALUATION

To further evaluate the generalizability of UniAG, we construct a composite dataset consisting of one industrial dataset, VisA, and two medical datasets, BrainMRI and HeadCT. We train *a single unified* model with UniAG on this dataset, and compare its generation quality and anomaly detection performance based on the generated data against existing *one-for-one* approaches, as shown in Tab. 5 and Tab. 6. Across nearly all anomaly categories in this composite dataset, UniAG achieves superior performance in both anomaly generation and detection, thereby demonstrating the effectiveness and generalizability of the proposed method. In particular, UniAG improves generation quality over existing methods by $(0.2, 0.06, 0.56)$ on the (*IS*↑, *IC-L*↑, *FID*↓) metrics, and further achieves gains of $(0.7, 0.3, 0.7, 0.7)$ on object-averaged ($AUROC_i$, $AP_i$, $AUROC_p$, $AP_p$) for anomaly detection and localization based on the generated data.

## 4.6 ABLATION STUDY

**Module Ablation** We design four UniAG variants: **V-0**: Simply pasting few-shot anomalous patches into random normal samples without local anomalous spatial-texture learning. **V-1**: Replacing our local deep copy–paste strategy with full-image anomaly inpainting Duan et al. (2023); Hu et al. (2024); Gui et al. (2024); Jin et al. (2025). **V-2**: Removing the proposed SCE module. **V-3**: Generation conditioned on textual embeddings Hu et al. (2024); Jin et al. (2025) instead of proposed $e_a$. We remove the SCE and AIA modules involved in $e_a$ and guide the generation in textual space while retaining the deep copy-paste generation strategy. Following Hu et al. (2024), we also report image-level anomaly classification accuracy $ACC_i$ with ResNet-34 He et al. (2016). Results in Tab. 7 and Fig. 7 highlight the following: (1) Naïve copy–paste (**V-0**) causes severe overfitting

Table 7: Component ablation.

| Variants | $ACC_i$ | $AUROC_p$ |
|---|---|---|
| V-0 | $42.7_{(\downarrow 25.8)}$ | $55.3_{(\downarrow 44.9)}$ |
| V-1 | $70.6_{(\downarrow 7.9)}$ | $88.5_{(\downarrow 1.7)}$ |
| V-2 | $69.6_{(\downarrow 8.9)}$ | $89.0_{(\downarrow 1.2)}$ |
| V-3 | $71.3_{(\downarrow 7.2)}$ | $88.4_{(\downarrow 1.8)}$ |
| UniAG | **78.5** | **90.2** |

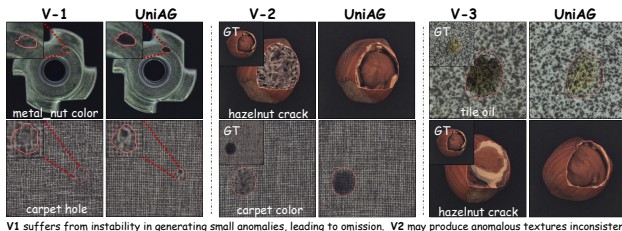

V1 suffers from instability in generating small anomalies, leading to omission. V2 may produce anomalous textures inconsistent with the target category. UniAG with $e_a$ generates results that are more faithful and closer to real samples than V3.

Figure 7: Visualizations of samples generated by UniAG and its variants using the same input.

and large performance drops. (2) Deep copy–paste improves local generation quality and mitigates anomaly fading, boosting $AUROC_p$ by **1.7** over overall anomaly samples generation (**V-1**). (3) The SCE module ensures category-consistent anomaly textures, raising $ACC_i$ by **8.9** over V-2. (4) Compared with textual embeddings (**V-3**), our proposed novel condition $e_a$, which integrates spatial and categorical cues, is more effective in producing realistic and diverse anomalies, leading to an improvement of **7.2** in $ACC_i$ and **1.8** in $AUROC_p$.

**Anomaly Fusion Ablation**   To blend locally generated anomaly patches into normal background images, we explore several strategies, including 1) *Poisson Blending* using implementation from a standard OpenCV function; 2) Additional fusion Autoencoder trained on pseudo data constructed from few-shot anomaly samples, where anomalous foreground regions are randomly augmented to introduce spatial and appearance inconsistencies between the foreground and background. 3) *Gaussian Feathering Blending* uses a Gaussian distribution to create smooth transitions, reducing harsh edges when stitching, compositing, or merging images. Anomaly localization performance comparisons with these blending methods are shown in Tab. 8. We find that the simple and efficient "Simple Pasting" approach yields better AD performance as it minimizes interference with both the normal background and the anomalous foreground, as shown in 8, ensuring precise alignment with the mask. Some may be concerned that the simple paste operation could introduce foreground–background boundary artifacts. While these may slightly affect the visual appearance of the generated data, their impact on supervised anomaly detection (segmentation or reconstruction-based)is minimal compared with maintaining consistency between the anomalous region and the mask. This is because anomaly detection inherently focuses on local texture inconsistencies, where even local gradient discontinuities naturally constitute valid anomalies that align with the mask. The single-pixel boundaries introduced by the operation are negligible relative to human-annotatable anomalous regions, as confirmed by prior works Zavrtanik et al. (2021a); Li et al. (2021).

Table 8: AD performance comparison on generated data using different anomaly fusion methods.

| Fusion methods | $AUROC_{pixel}$ | $AP_{pixel}$ |
|---|---|---|
| Poisson Blending | $87.5_{(\downarrow 2.7)}$ | $66.9_{(\downarrow 3.0)}$ |
| Fusion Autoencoder | $88.9_{(\downarrow 1.3)}$ | $69.8_{(\downarrow 0.1)}$ |
| Gaussian Feathering | $89.3_{(\downarrow 0.9)}$ | $70.1_{(\uparrow 0.2)}$ |
| UniAG (Direct Pasting) | **90.2** | **69.9** |



Figure 8: Visualizations of samples generated by different blending methods. Poisson blending may distort the background, Gaussian feathering smooths edges but may cause inconsistencies, and the fusion autoencoder cannot fully fix boundary issues while adding computational overhead.

## 5 CONCLUSION

In this paper, we propose UniAG, a diffusion-based few-shot anomaly generation framework that leverages a deep copy–paste strategy to learn and generate realistic, diverse multi-class anomalies efficiently. Built upon this, the Spatial-Texture Alignment Diffusion Model (STA-DM) incorporates essential spatial and category information as generation conditions, further enhancing the fidelity and diversity of generated anomalies. Extensive experiments on the commonly used dataset demonstrate that UniAG not only produces high-quality anomalous samples but also consistently improves performance across downstream tasks, including anomaly detection, localization, and classification.

## ETHICS STATEMENT

This work focuses on algorithmic improvements and experimental evaluation on publicly available datasets. No sensitive data, human subjects, or personally identifiable information are involved. We do not anticipate any potential ethical concerns related to this research.

## REPRODUCIBILITY STATEMENT

To ensure the reproducibility and completeness of this paper, we have included an Appendix consisting of four main sections. In Appendix C, we introduce the details of the datasets used in our experiments and provide additional comparative results on these datasets. Appendix D presents more implementation details of UniAG. In Appendix E, we provide comparisons with unsupervised generation methods, implementation details of input mask augmentation, and discussions on comparison with existing methods. We report efficiency analysis in F and more visualizations in Appendix G. Our code will be made publicly accessible once the paper is accepted.

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

## APPENDIX

## A  THE USE OF LARGE LANGUAGE MODELS (LLMS)

This work involves the assistance of large language models (LLMs) for text polishing and grammar checking only. No LLMs were used for developing the core methodology, conducting experiments, or analyzing results.

## B  MULTI-CATEGORY ANOMALY GENERATION DETAILS

We compare in Fig. 9 the difference in setting between UniAG's multi-category anomaly generation and the one-to-one generation approach adopted by existing methods. The multi-category anomaly generation offers the following advantages:

(1) Enhancing training efficiency: Unlike existing methods that require training a separate model for each anomaly type on MVTec AD (a total of 72 models), UniAG trains only a single unified model. This significantly improves training efficiency; as shown in Fig. 1 in the main text, UniAG reduces the training time by 95.4% compared to DualAnoDiff.

(2) Improve practical generation: Using a single generative model facilitates efficient deployment. At the same time, UniAG allows users to specify anomaly categories and regions, providing interactive and customizable generation options that enrich

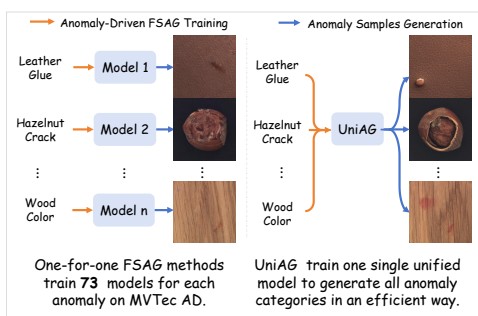

Figure 9: Comparison between UniAG's multi-category anomaly generation (AG) and existing one-to-one AG models

application scenarios and support the output of any possible anomaly patterns with arbitrary distributions for continuously improving detection models.

(3) Most importantly, under a multi-category anomaly generation setup, UniAG achieves superior performance compared to existing one-for-one anomaly generation methods, both in terms of

anomaly generation quality and subsequent anomaly detection evaluation. With the same amount of generated data (100 samples), UniAG outperforms existing one-model-for-one-anomaly methods on the MVTec AD dataset, achieving an improvement of $(0.13, 0.02, 1.08)$ on (*IS*, *IC-L*, *FID*) in anomaly generation metrics and an increase of $(2.3, 2.5)$ on ($AUROC_i$, $AUROC_p$) in anomaly detection and localization performance.

## C    DATASET AND ADDITIONAL EVALUATIONS

Following Duan et al. (2023); Hu et al. (2024); Gui et al. (2024); Jin et al. (2025), we conduct FSAG experiments solely on this dataset due to its sufficient comprehensiveness. MVTec AD contains a substantially larger number of objects (15) and anomaly categories (73) than other datasets, such as VisA (12 anomaly categories), making it a comprehensive benchmark for generation evaluation. To further evaluate the generalizability of UniAG, we construct a composite dataset consisting of a subset of VisA (6 objects with 6 anomaly categories) and two medical datasets, BrainMRI and HeadCT. We train *a single unified* model with UniAG on this dataset, and compare its generation quality and anomaly detection performance based on the generated data against existing *one-for-one* approaches, as shown in Tab. 5 and Tab. 6. Across nearly all anomaly categories in this composite dataset, UniAG achieves superior performance in both anomaly generation and detection, thereby demonstrating the effectiveness and generalizability of the proposed method. In particular, UniAG improves generation quality over existing methods by $(0.2, 0.08, 0.56)$ on the (*IS* $\uparrow$, *IC-L* $\uparrow$, *FID* $\downarrow$) metrics, and further achieves gains of $(0.7, 0.5, 0.7, 0.7)$ on object-averaged ($AUROC_i$, $AP_i$, $AUROC_p$, $AP_p$) for anomaly detection and localization based on the generated data.

**MVTec AD** Bergmann et al. (2019) A widely used industrial inspection dataset comprising **5354** images (**4096** normal and **1258** anomalous). The image resolution has a shorter side ranging from **700** to **1024**. The dataset contains **15** object categories, encompassing a total of **73** anomaly types.

**VisA Zou et al. (2022)**. A multi-domain dataset consists of 10,821 images (9,621 normal and 1,200 anomalous samples) in total, which covers 12 objects in 3 domains, with each object containing one specific anomaly type. Its relevant domains including printed circuit boards (PCBs) from the electronics sector and food items such as macaroni and fryum.

**BrainMRI Salehi et al. (2021)**, a single-category medical anomaly dataset consisting of 98 normal MRI images and 155 with tumors.

**HeadCT Salehi et al. (2021)**, a single-category medical anomaly dataset containing 25 normal images and 100 images with hemorrhage, where each CT image comes from a different person.

## D    IMPLEMENTATION DETAILS

All images in MVTec-AD are resized to 256 ×256. We utilize the KL method as the Auto-encoder and fine-tune the model on the few-shot datasets before training the denoising network as in He et al. (2024). We train for 1000 epochs with a batch size of 12. Adam optimizer Kingma & Ba (2014) with a learning rate of $1e - 5$ is set. For anomaly detection, the anomaly score of the image is the maximum value of the anomaly localization score following the implementation of Hu et al. (2024). During inference, we use DDIM Song et al. (2021) as the sampler with 10 steps by default. We implement all experiments on a single NVIDIA RTX 3090 GPU with 24G memory.

## E    DISCUSSION

### E.1    CROSS-CATEGORY ANOMALY TRANSFERRING

Cross-category anomaly transferring is a promising approach for zero-shot unseen anomaly generation Shi et al. (2024); Sun et al. (2025). However, the latest method, AnomalyAny Sun et al. (2025), shows AD performance (image-AUROC, pixel-AUROC) $(1.0, 2.0) \downarrow$ lower than UniAG as reported in their official results. We argue that relying only on cross-category anomaly transferring has inherent limitations: 1) it heavily relies on contextual priors, making it unclear which categories can serve as valid sources; 2) it struggles to handles domain-specific rare anomalies with no meaningful

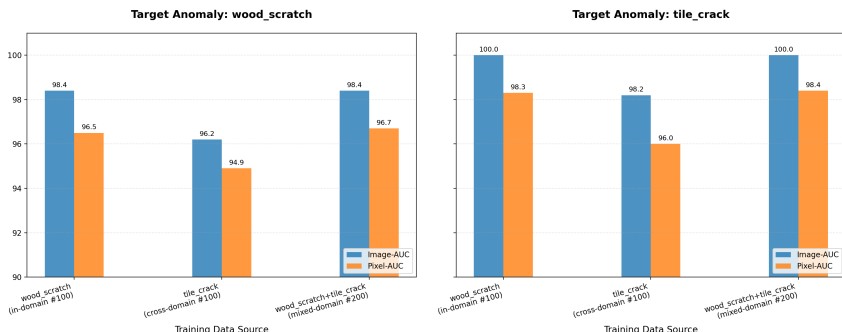

Figure 10: Comparison of the impact of in-domain, cross-domain, and mixed-domain anomaly transferring generation on anomaly detection performance.

Table 9: Comparison of different methods in anomaly region mask generation/augmentation along with their advantages and drawbacks. Our UniAG adopts a model-free spatial augmentation approach, which reduces computational overhead while maintaining mask diversity. Similar to existing methods Hu et al. (2024), we use object locations as priors for plausible anomaly regions.

| Methods | Mask Generation/Augmentation | Advantages | Drawbacks |
|---|---|---|---|
| DFMGAN | Output from an extra mask generation branch, along with RGB branch. | Partially enhanced their diversity. Improved the spatial plausibility of masks. | Increases training and inference overhead, and mask acquisition is coupled with anomaly generation, which hinders mask optimization and limits diversity. |
| AnomalyDiffusion SeaS | Using an additional textual-inversion model for mask generation. | Enhanced mask diversity | Requires additional training and inference overhead, which may produce unreasonable masks (e.g., too small in area) that need to be filtered based on predefined rules. Requires target foreground location in normal samples to ensure the validity of pixels in the augmented masks. |
| AnoGen | 100 pre-defined boxes as coarse masks. These boxes are evaluated based on their overlap with the target foreground and manually inspected, and then randomly selected as region mask for generation. | Enhanced mask diversity. Simple, efficient, and easy to use | Both realism and diversity are insufficient, and excessive randomness leads to a high cost for mask inspection. Requires target foreground location in normal samples to ensure the validity of pixels in the augmented masks. |
| DualAnoDiff | A segmentation model (U²-Net) is applied to the simultaneously generated anomalous foreground image to obtain the corresponding mask. | Partially enhanced their diversity. Improved the spatial plausibility of masks. | Mask acquisition is coupled with anomaly generation, limiting scalability and accuracy. Extra segmentation inference is needed, and generation errors can lead to masks misaligned with actual anomaly textures. |
| UniAG | Apply random spatial augmentations to masks in few-shot anomalous samples | Enhanced mask diversity. Simple, efficient, easy to use, requires no manual or rule-based filtering. | Requires target foreground location in normal samples to ensure the validity of pixels in the augmented masks. |

cross-category equivalents; and 2) without real anomaly guidance, generated samples, though visually plausible, exhibit a domain gap with the real test anomalies, offering limited training utility and sub-optimal AD performance.

We further investigate the effect of using in-domain anomaly textures versus cross-domain transferred textures (*e.g.*, wood scratch → tile crack, tile crack → wood scratch) on anomaly detection performance. Under identical experimental settings (UniAG with U-Net, single-anomaly setting), we evaluate the AD performance as illustrated in Fig. 10. While cross-category texture transfer yields reasonable detection performance, it remains inferior to the in-domain training adopted by UniAG, which uses 4-shot anomaly prompts to generate samples that more closely resemble real test anomalies. Nevertheless, cross-domain transfers can serve as a valuable complement to UniAG by enriching texture diversity and potentially further enhancing anomaly localization when combined with UniAG's realistically generated samples.

### E.2 SENSITIVE TO IMPERFECT MASK QUALITY.

In FSAG, the mask defines the anomaly pattern and also drives subsequent (image, mask) generation. Thus, all FSAG methods depend on accurate anomaly masks in principle. As with prior works Hu et al. (2024); Dai et al. (2024); Jin et al. (2025), we follow the standard assumption that anomaly datasets provide reasonably aligned masks in the AD datasets.

We investigate the anomaly generation and detection performance under imperfect few-shot masks with mislabeling. On the first three alphabetically sorted categories (*i.e.*, bottle, cable, capsule) from the MVTec AD dataset, we use the interior of the original mask's bounding box as an inconsistency mask (b-box). Then, using the same experimental settings as Tab.2, we compared the anomaly detection performance against DualAnoDiff Jin et al. (2025). Comparison of detection performance evaluated with ground truth masks is shown in Tab.10. UniAG shows only a minor $(0.3, 0.1)$ under

Table 10: Performance comparison using imperfect masks with inconsistency.

| Mask | Image-AUC | Pixel-AUC |
|------|-----------|-----------|
| DualAnoDiff (b-box) | 91.1 ($\downarrow$ 1.9) | 80.6 ($\downarrow$ 1.5) |
| DualAnoDiff (gt mask) | 93.0 | 82.1 |
| UniAG (b-box) | 96.5 ($\downarrow$ 0.3) | 86.9 ($\downarrow$ 0.1) |
| UniAG (gt mask) | 96.8 | 87.0 |

inconsistent masks, compared to the performance drop $(1.9, 1.5)$ of DualAnoDiff, demonstrating lower sensitivity. This is because UniAG uses spatial–category aligned embeddings to explicitly constrain anomaly generation within the given mask, providing an upper bound on mask error. In contrast, DualAnoDiff generates local anomalies globally and then conducts foreground segmentation, which may amplify mask misalignment due to anomaly content confusion and the lack of an error upper bound.

### E.3 MASK AUGMENTATION: COMPARATIVE ANALYSIS AND DETAILS.

Reasonable and diverse masks are important prior to anomaly generation. However, under a few-shot setting, determining plausible and varied anomaly locations is highly challenging due to several factors: (1) anomaly locations exhibit strong structural priors and, compared to anomaly textures, can have more diverse plausible configurations; (2) imaging jitter or misalignment introduces uncertainty in the positions of foreground objects, making it harder to determine anomaly locations; (3) Irregular objects in shape may further complicating the alignment of reasonable anomaly positions.

To achieve diverse and plausible anomaly locations, we compare our approach with existing methods, as shown in Tab. 9. Similar to Anodiff and AnoGen, we adopt a generation-decoupled approach, which avoids the influence of full-image generation and explicitly enhances mask diversity. We employ a simple spatial-variation-based mask augmentation, which is more convenient and efficient. To ensure that the augmented anomalous masks lie within the target foreground region, we also use the foreground mask to provide spatial information. Specifically, the intersection between the augmented anomalous mask and the object foreground mask is used as the final input to the model.

Specifically, we apply the following spatial transformations to augment anomaly region masks of few-shot anomalous data to enhance mask diversity:

```python
import albumentations as A
[A.RandomRotate90(), A.Flip(),
A.Transpose(),
A.OpticalDistortion(p=1.0, distort_limit=1.0),
A.ShiftScaleRotate(shift_limit=0.0625,  scale_limit=0.2, rotate_limit=45, p=0.2),
A.OneOf([
    A.OpticalDistortion(p=0.3),
    A.GridDistortion(p=.1),
    A.IAAPiecewiseAffine(p=0.3),
], p=0.2),
]
```

After generating the anomalous patches, we apply the following appearance transformations to further enhance the diversity of anomaly textures:

```python
[A.OneOf([
    A.IAAAdditiveGaussianNoise(),
    A.GaussNoise(),
], p=0.2),
A.OneOf([
    A.MotionBlur(p=.2),
    A.MedianBlur(blur_limit=3, p=0.1),
    A.Blur(blur_limit=3, p=0.1),
], p=0.2),
A.OneOf([
    A.CLAHE(clip_limit=2),
    A.IAASharpen(),
    A.IAAEmboss(),
    A.RandomBrightnessContrast(),
], p=0.3),
A.HueSaturationValue(p=0.3)]
```

### E.4 TIGHTEN NORMALITY VS. ANOMALY SIMULATION.

Unsupervised AD methods Zavrtanik et al. (2021a); Li et al. (2021) generate simple anomalies online with only normal samples, which require a large amount of data to model the normal boundary accurately. In contrast, FSAG methods Duan et al. (2023); Hu et al. (2024); Gui et al. (2024); Jin et al. (2025) simulate real and diverse anomalies with few-shot anomalous samples. Realistic and diverse anomaly generation provides three main advantages:

1. *Improved anomaly localization performance.* FSAG generates spatially aligned, realistic, and diverse anomalous samples along with corresponding mask pairs. This allows better anomaly localization even when detection performance is comparable.

2. *Data efficiency.* Since actual anomalous patterns are much rarer than normal patterns, focusing on anomalies enables detection models to achieve superior performance with fewer training samples, significantly reducing generation and training costs.

3. *Support for diverse anomaly recognition tasks.* FSAG-generated data can be used not only for detection but also for anomaly classification, anomaly description, and other downstream tasks.

4. *Enhanced interpretability.* Supervised anomaly detection trained with FSAG data provides clearer explanations and facilitates optimization for both false positives and false negatives.

Our proposed UniAG falls under the category of (FSAG) methods, and we compare UniAG with DRAEM on pixel-level AD performance as shown in Tab. 11. Under the same amount of data and using the same detection model, UniAG achieves significantly better performance, regardless of whether segmentation-based or reconstruction-based methods are employed.

Table 11: Comparison between UniAG and SOTA unsupervised normal-boundary-modeling anomaly generation methods.

| | DRAEM origin | DRAEM w. UNet | UniAG w. UNet | DRAEM w. FSN | UniAG w. FSN |
|---|---|---|---|---|---|
| $AUROC_{pixel}$ | 97.3 | 76.2 | 90.2 | 97.5 | **99.2** |
| $AP_{pixel}$ | 68.4 | 46.4 | 69.9 | 69.2 | **81.0** |

Table 12: UniAG improves generation efficiency by using the minimal (10) DDIM steps.

| | AnomalyDiffusion | AnoGen | DualAnoDiff | UniAG |
|---|---|---|---|---|
| DDIM steps | 200 | 50 | 100 | **10** |

### E.5 GENERATION EFFICIENCY.

With numerous anomaly categories, supervised AD models typically demand large amounts of training data. Therefore, improving generation efficiency is crucial for practical applications.

Our proposed UniAG enables more efficient anomaly generation, primarily reflected in the following aspects: (1) It requires fewer inference DDIM steps, as shown in Tab. 12, significantly reducing inference time; (2) It does not rely on manually specified foreground mask priors, which are required in methods like AnoGen; (3) It avoids the mask generation inference process used in AnomalyDiffusion, thereby simplifying the generation pipeline; (4) By directly using spatially enhanced masks as input, it produces paired anomalous data without the need for an additional segmentation on generated foregrounds, as required in DFMGAN and DualAnoDiff.

## F    LIMITATION

Similar to existing FSAG methods, reasonably expanding the spatial region masks of specific anomalies from few-shot visual examples remains challenging, particularly in achieving both plausibility and diversity of anomalous regions. This difficulty stems from an inherent contradiction: the need for a strong prior on anomaly location versus the lack of sufficient information in a few-shot

setting. Future work could explore the integration of expert knowledge and rule-based approaches to alleviate this limitation.

## G   MORE VISUALIZATION

We present additional visualizations of anomaly samples generated by UniAG across all 15 object categories in the MVTec AD dataset in Fig. 11 and Fig. 12.

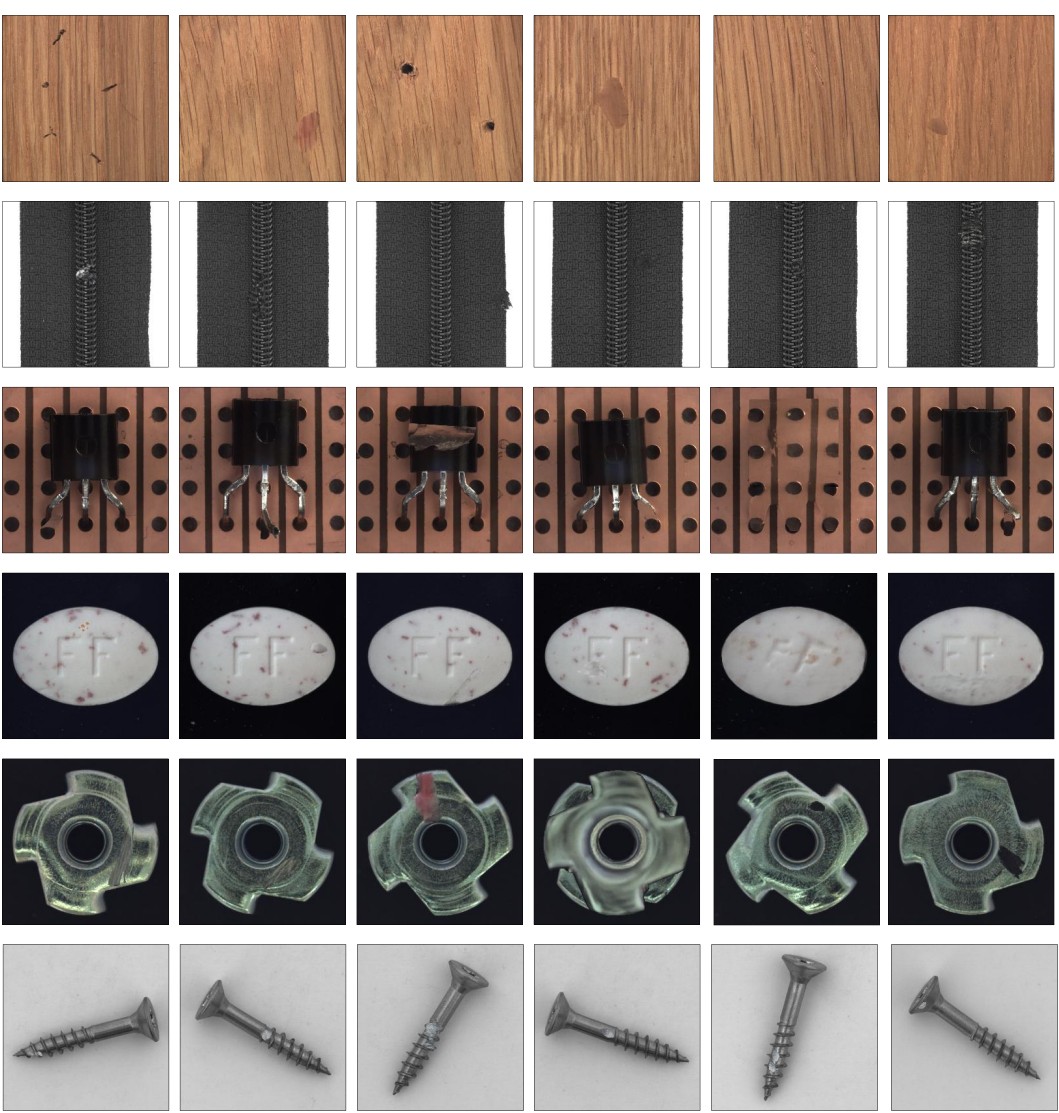

Figure 11: More visualization of generated samples of UniAG.

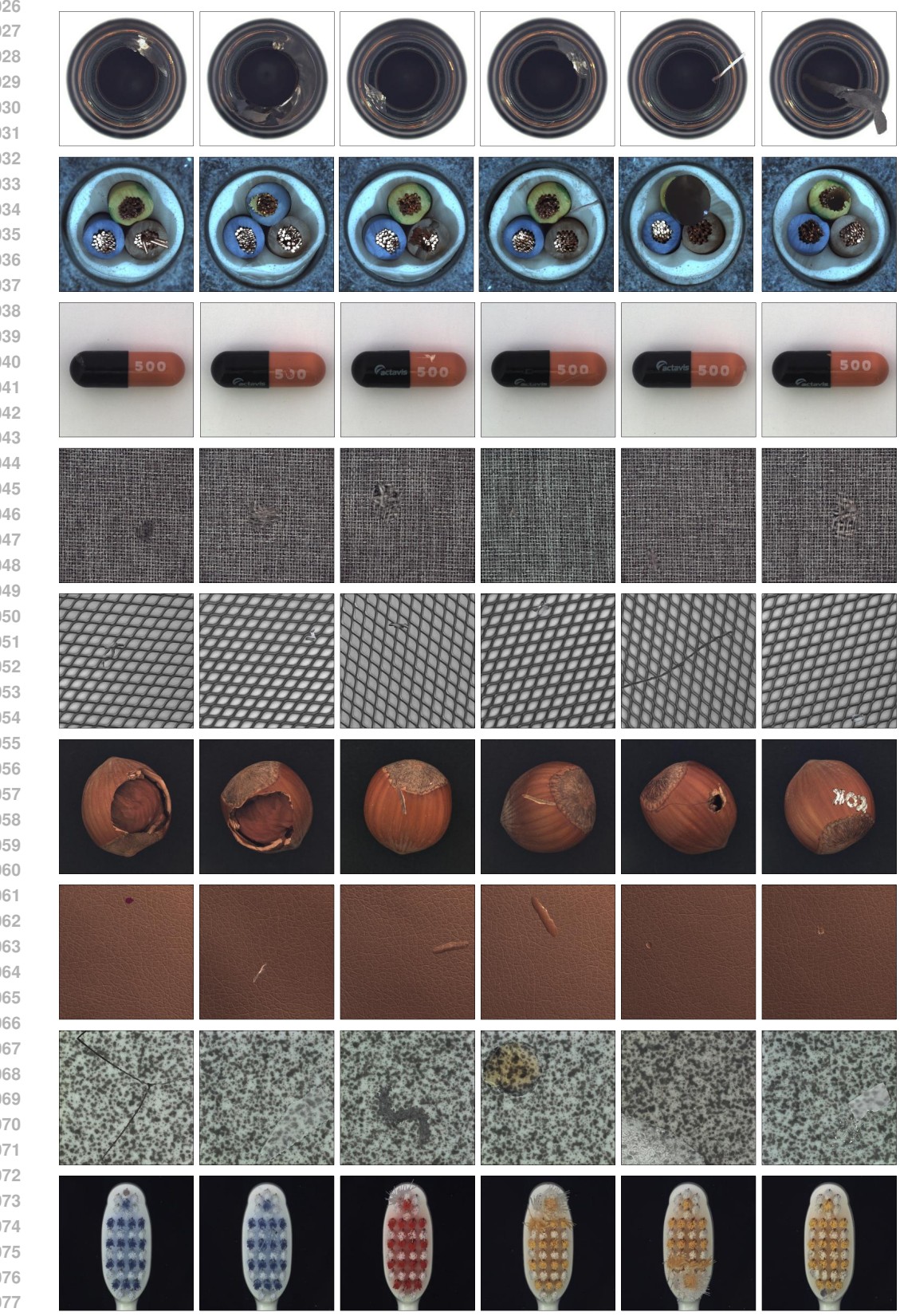

Figure 12: More visualization of generated samples of UniAG.

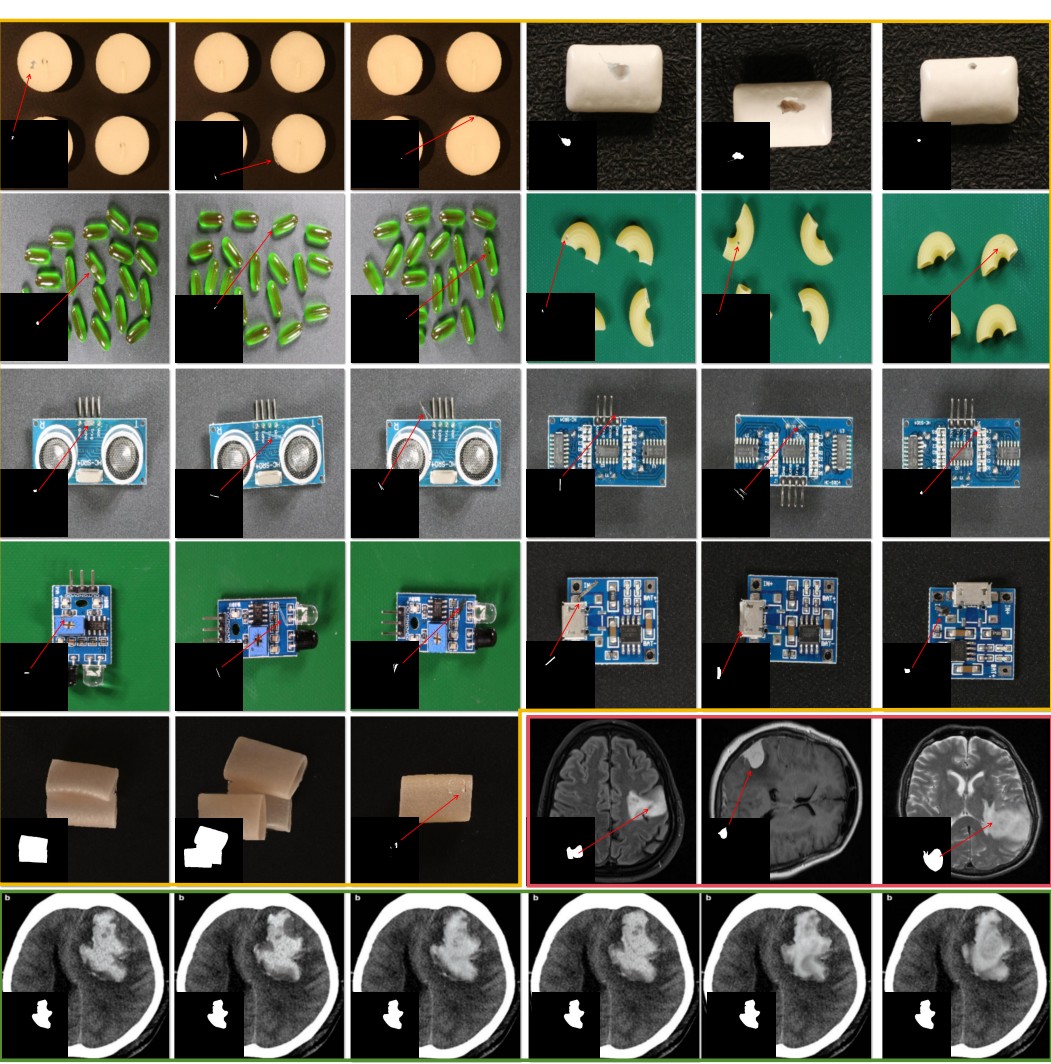

Figure 13: More visualization of generated samples on a composite dataset consisting of VisA, BrainMRI, and HeadCT. We used the same background on HeadCT samples to verify the diversity of generated anomalous textures with fixed location and anomalous category.

