# OpenReview forum: "UniAG: Unified Anomaly Generation via Local Spatial-Texture Alignment Diffusion Model"
_ICLR.cc/2026/Conference — ICLR 2026 Conference Withdrawn Submission_

### Official Review · Reviewer_mHJL · 2025-10-27

**Soundness:** 2
**Presentation:** 2
**Contribution:** 2
**Rating:** 2
**Confidence:** 5

**Summary:**

This paper proposes UniAG, a unified diffusion-based model for few-shot anomaly generation. The authors design a Spatial–Texture Alignment Diffusion Model (STA-DM) that introduces a spatial-category embedding and an anomaly injection adapter to unify the generation of multi-class anomalies within a single model. The authors claim that this approach improves training efficiency, generation quality, and downstream anomaly detection performance compared to existing category-specific models.

**Strengths:**

1. The writing is clear, and the figures effectively illustrate the performance.
2. The idea of decoupling texture generation from background modeling via a localized strategy is intuitively appealing.

**Weaknesses:**

1.	The motivation of this paper is problematic. Although the scarcity of anomaly samples and the diversity of anomaly patterns pose a challenge to anomaly detection as the authors declared in the introduction, many unsupervised anomaly detection methods, especially reconstruction-based methods such as [a] and [b], achieve higher scores using the same dataset (MVTec AD) compared to the results reported in this paper. These methods only use normal samples to train the model and do not need any synthetic data to augment the existing training data. In this case, it is difficult to see any meaningful value to generate such anomaly images, especially for industrial applications.

[a] One Dinomaly2 Detect Them All: A Unified Framework for Full-Spectrum Unsupervised Anomaly Detection (an extension of CVPR2025)

[b] INP-Former++: Advancing Universal Anomaly Detection via Intrinsic Normal Prototypes and Residual Learning (an extension of CVPR2025)

2.	The proposed UniAG framework largely integrates existing components from prior diffusion-based anomaly generation and inpainting literature. The ‘deep copy–paste’ strategy resembles prior local inpainting and patch-based augmentation methods. The ‘spatial–category embedding’ and ‘adapter’ modules are minor variations of established conditional diffusion techniques (e.g., ControlNet, structure-guided diffusion). The paper does not introduce a new learning principle or a fundamentally novel diffusion mechanism. Therefore, the contribution is incremental and mainly engineering-oriented. In my opinion, the paper primarily focuses on empirical gains on one benchmark without offering new insights into anomaly generation or diffusion theory. The work reads more like an application paper suited for an applied vision venue rather than a top-tier learning conference like ICLR.
3.	The paper lacks formal analysis or theoretical justification of why STA-DM improves over text-conditioned diffusion. The model description essentially reproduces standard conditional diffusion objectives with added embeddings.
4.	The comparison methods are limited and many current SOTA methods are not compared especially published in 2025.
5.	In the main text, the experiments are almost entirely limited to MVTec AD, a single industrial dataset. This is not convincing to support the main contribution of this paper because it is a very easy dataset for anomaly detection. Furthermore, only 4-shot is analyzed in this paper, lacking the validation of generalization of other numbers of few-shot.
6.	The architectural contributions are not clearly distinguished from existing conditional diffusion models such as ControlNet [c].

[c] Adding conditional control to text-to-image diffusion models. (CVPR2023)

**Questions:**

Except for the major concerns presented above in the Weaknesses, I have another two additional questions:

1.	In Equation (4) and Section 3.1, the latent variable z_a^t and the binary mask \hat{m} lie in different spaces (latent vs. pixel), with different spatial resolutions and channel dimensions. How is the mask’s spatial resolution and channel depth adapted to enable the described element-wise multiplication with the latent tensor?

2.	Regarding Figure 4, what is the exact category identifier used during training and inference for the bottle example? If "hazelnut cut" is used for a bottle, how does the model prevent generating semantically incorrect textures, and what is the rationale behind this design?

---

> ### Author Response · Authors · 2025-11-20
> **Response to Reviewer mHJL.(Part 1)**
>
> Thank you for your review and feedback. We address your concerns one by one as follows.
> ## 1. Clarification on the Motivation of FSAG
> We are not the first to introduce FSAG, but focus on analyzing and addressing limitations of existing FSAG methods [1–6]. The motivation for FSAG is well established in previous works and claimed in submission lines 46–47. Benefiting from the full utilization of few-shot task-specific anomaly references, FSAG generate anomaly textures similar to real data, which is beneficial for supervised training and reduces the training-test domain gap to achieve better performance than unsupervised[8] or zero-shot[9] anoamly generation methods. FSAG-based supervised anomaly detection provides better anomaly localization and supports fine-grained anomaly classification. Tab.3,4 compare anomaly localization performance with state-of-the-art unsupervised methods and report anomaly classification accuracy. Unsupervised methods show suboptimal localization and cannot distinguish anomaly categories.
> ## 2.Incremental Contribution and Engineering-oriented
> We clarify that UniAG is not an incremental engineering work, but a systematic study addressing four key limitations of existing FSAG methods in realism, diversity, and stability (lines 48–71, 141–149; Fig. 2). To this end, UniAG introduces:
> 1. A locally-enhanced deep copy–paste strategy for realistic, diverse, and stable anomaly generation, especially for small anomalies.
> 2. A spatial–texture alignment diffusion model with spatial–category embedding, enabling controllable generation at both position and category levels, and mitigating the inefficiency of one-model-per-anomaly training.
>
> In fact, we rigorously follow prior experimental settings and report additional results on additional datasets in the supplementary material to demonstrate generality. This work also aligns with ICLR’s scope, addressing applications in computer vision, audio, language, and other modalities.
> ## 3.Advantages of STA-DM over Text-conditioned Diffusion Models
> Text-conditioned diffusion models learn only implicit anomaly semantics, whereas STA-DM uses anomaly embeddings ($e_a$) to provide distinguishable category and spatial region conditioning.
>
> STA-DM uses explicit spatial–category cues to guide realistic texture generation, while text-conditioned diffusion relies on implicitly learned embeddings, which need to focus on both texture and position generation and do not support multi-class generation. In ablation experiments (V3, rows 467–470) under identical conditions, STA-DM improved classification accuracy by 7.2 and pixel-level AUROC by 1.8 (Table 5, Fig. 7), confirming its effectiveness.
> ## 4.Comparison with FSAG methods in 2025
> We compare our method with the latest FSAG approach, DualAnoDiff [4] (CVPR 2025), as shown in the Tab.1,2,3,4 and Fig.2,5,6 in the paper. At submission, some new FSAG methods were not yet released. We report comparisons with Fast [5] and SeaS [6] which are fully released recently, reporting results for anomaly generation and detection (100 samples) below.
> | Methods | Publish |FID ⬇️|IS⬆️|IC-LPIPS⬆️|Image-AUC |Pixel-AUC |
> |----------|----------|----------|----------|----------|----------|----------|
> | Fast[5] | NeurIPS2025 |0.65|2.20|0.45|92.1| 86.2  |
> | SeaS[6] | ICCV2025  | 2.43 |2.41 |0.52|94.8|78.5|
> | UniAG| Submission | 0.53|2.58|0.51|95.6 | 90.2|
>
> The comparison against the latest open-source methods will be added to the revision.
> ## 5.Single Dataset and 4-shot Settings
> As stated in lines 359–361, rigorously following current FSAG works [1–5], we conduct few-shot experiments primarily on comprehensive MVTec AD.  To further demonstrate generality, we report results on a mixed dataset (VisA+medical) in the *Supp.Sec.C*. We adopt a 4-shot setting to reflect real-world anomaly scarcity, more realistic than prior methods using roughly one-third of anomaly data per category [2–4].
> ## 6.Distinction from ControlNet of AIA Module
> As described in lines 345–350, AIA borrows ControlNet’s conditional fusion structure to integrate the spatial–category embedding ($e_a$, Eq. 7), with key differences in the generation condition and training objective (Eq. 9), focusing on realistic texture synthesis within masked regions. The core contributions of our work are the deep copy–paste strategy and spatial–category alignment diffusion model (STA-DM); the AIA module serves only as a conditional fusion component, not a primary innovation.

---

> ### Author Response · Authors · 2025-11-20
> **Response to Reviewer mHJL.(Part 2)**
>
> ## 7. Dimension issues in Eq. (4)
> In Eq. (4), the latent space is (B, 4, 32, 32), and the mask (B, 1, 256, 256) is resized to (B, 1, 32, 32) before multiplication. We acknowledge this was briefly described and will clarify in the revision.
> ## 8.Anomaly identifier and multi-class generation.
> As stated in lines 238–239, each anomaly is assigned a unique id ($\mathcal{A}_n$) using its global category index for training and inference. During multi-class generation, UniAG takes both mask and category id as inputs, providing spatial and categorical information to generate matching textures. Ablation results (Fig. 7, V2) show that omitting the category input can produce unrealistic or irrelevant textures.
>
> [1] Duan et al. DFMGAN. AAAI2023.
>
> [2] Hu et al. Anomalydiffusion. AAAI2024.
>
> [3] Guan et al. AnoGen. ECCV2024.
>
> [4] Jin et al. DualAnoDiff. CVPR2025.
>
> [5] Xu et al. FAST. NeurIPS2025.
>
> [6] Dai et al. SeaS. ICCV2025.
>
> [7] Zhang et al. PRN. CVPR2023.
>
> [8] Zhang et al. Realnet. CVPR2024.
>
> [9] Sun et al. AnomalyAny. ICCV2025.

---

### Official Review · Reviewer_buaZ · 2025-10-29

**Soundness:** 3
**Presentation:** 4
**Contribution:** 3
**Rating:** 6
**Confidence:** 4

**Summary:**

The paper proposes UniAG, a novel framework for Few-Shot Anomaly Generation (FSAG). Its core innovation is a unified model that can generate realistic and diverse anomalies for multiple categories, addressing a key inefficiency in prior work that required training a separate model for each anomaly type. The method combines a "deep copy-paste" strategy with a custom Spatial-Texture Alignment Diffusion Model (STA-DM) that uses explicit spatial-category embeddings instead of ambiguous text prompts. The results on the MVTec AD benchmark are impressive, demonstrating state-of-the-art performance in both generation quality and downstream anomaly detection tasks.

**Strengths:**

1. Motivation: The paper clearly identifies the limitations of existing FSAG methods: The need to train one model per anomaly type is computationally expensive and impractical for deployment.
2. Well-Motivated Methodology: Replacing text embeddings with a learned spatial-category embedding (e_a) is a key contribution. It provides more precise control over the generation process, leading to higher fidelity.
3. Experiments: Studied datasets are thourough. The ablation studies are thourough to validate the contribution of each component of the method. The scalability analysis (Figure 6) effectively demonstrates the practical advantages of UniAG as the number of categories and the scale of generated data increase.
4. Clarity and Reproducibility: The paper is well-structured and clearly written. The appendices provide substantial additional detail on datasets, implementation, mask augmentation, and fusion strategies, which is commendable for reproducibility.

**Weaknesses:**

1. Simplicity of "Direct Pasting": The paper finds that the simplest blending strategy, "direct pasting," performs best for downstream detection tasks. While this is efficient, it can create sharp, unrealistic edges between the anomalous patch and the normal background. The authors state this has "little effect", but this could be a drawback for anomaly types where the boundary itself is a key feature (e.g., subtle gradients or stains)
2. Clarification on contribution: The claim in the introduction to be the "first unified model to support multi-class anomaly generation"  requires more precise qualification. This claim is very strong, and other recent works have also proposed unified "one-for-all" generation models, such as [1] (unsupervised synthesis method) and [2][3] (zero-shot method using pre-trained diffusion models). While UniAG's contribution is unique, it should be clearly and explicitly limited to the Few-shot Anomaly Generation (FSAG) setting, where a single model is trained from a small set of real anomaly examples. This distinction from unsupervised or zero-shot paradigms should be clarified.
3. More visual results: While the paper provides extensive quantitative results on the composite dataset (VisA, BrainMRI, HeadCT) in Appendix C , it lacks corresponding visual examples. To fully substantiate the claims of generalizability, adding a few qualitative examples of generated anomalies from the VisA and medical datasets would strengthen the results.

[1] Zhang, Ximiao, Min Xu, and Xiuzhuang Zhou. "Realnet: A feature selection network with realistic synthetic anomaly for anomaly detection." Proceedings of the IEEE/CVF conference on computer vision and pattern recognition. 2024.
[2] Sun, Han, et al. "Unseen Visual Anomaly Generation." Proceedings of the Computer Vision and Pattern Recognition Conference. 2025.
[3] He, Shidan, et al. "Anomalycontrol: Learning cross-modal semantic features for controllable anomaly synthesis." arXiv preprint arXiv:2412.06510 (2024).

**Questions:**

1. A key question is: how does the model perform when a new anomaly category is introduced? Can the model be incrementally fine-tuned, or does it require retraining from scratch on all categories? A discussion on this "incremental learning" aspect would strengthen the paper's contribution to a truly deployment-friendly system. If retraining is required, the efficiency benefit of the unified model is significantly diminished over time, as it would be more costly than simply training one new light-weighted model for the one new class. A discussion on this incremental learning capability is a missing, yet critical, piece of the evaluation.
2. The paper tests UniAG's generalizability on a composite dataset that includes BrainMRI and HeadCT images. Are the compared works on BrainMRI and HeadCT the SOTA methods in the medical field?

---

> ### Author Response · Authors · 2025-11-20
> **Response to Reviewer buaZ. (Part 1)**
>
> We greatly appreciate your thoughtful review and feedback. Our responses to your concerns are as follows.
> ## 1. Simple Pasting Limits Boundary-Sensitive Anomaly
> Unlike semantic recognition tasks, anomaly detection focuses on identifying **local texture inconsistencies**. In this context, **local gradient discontinuities also constitute a valid anomaly pattern** and naturally **align with the anomaly mask**. Importantly, the proportion of **single-pixel boundary introduced by simple paste operation is negligible** compared with the area of the **human-annotatable anomalous texture**, thus having **little effect** on subsequent supervised anomaly segmentation or reconstruction-based anomaly detection methods. This has already been confirmed in prior works which adopt the same "Simple Paste" blending to construct anomalies[1-4].
>
> Regarding the potential mismatch between the pasted anomalous foreground and the normal background (e.g., lighting or imaging inconsistencies), such discrepancies are themselves consistent with the definition of anomaly patterns. In fact, we further apply additional appearance augmentations—such as color and brightness changes, Gaussian noise, and blur—to **increase anomaly diversity** and enhance the **coverage** of the generated patterns.
>
> In *Supp.Sec.E.Tab.11*, we compared several strategies for removing clear boundaries and additionally include a new **Gaussian-Feathered Boundary Blending** as below. The results show that these techniques tend to **weaken real anomalous patterns**, causing **uncertain inconsistencies between the anomaly regions and the paired mask**, and ultimately leading to **suboptimal anomaly detection performance**.
>
> | Fusion Methods | Pixel-AUROC |Pixel-AP |
> |----------|----------|----------|
> | Poisson Blending | 87.5 ($\downarrow2.7$)            | 66.9($\downarrow3.0$) |
> | Fusion Autoencoder  | 88.9($\downarrow1.3$)             | 69.8($\downarrow0.1$) |
> | Gaussian Feathering Blending    | 89.3($\downarrow0.9$)       | 70.1($\uparrow0.2$) |
> | Simple Pasting    | **90.2**       | **69.9** |
>
>
> ## 2. Clarification on Contribution.
> In lines 91–93, we claim that "UniAG is the first unified model to **support multi-class anomaly generation** and **faithfully simulate real anomalous samples**". We demonstrate through comparisons that this claim is well-supported:
>
> 1. Unsupervised anomaly generation methods, such as DRAEM and RealNet, also use a single model to generate anomalies across all categories. However, due to the lack of guidance from real anomaly references, these methods often rely on non-local homogeneous textures or additional noise to simulate anomalies. They require synthesizing a large number of abnormal patterns in order to model the boundaries of the normal distribution. In contrast, UniAG, as an FSAG method, leverages task-specific anomaly information to generate samples that closely resemble real anomalous references, achieving superior performance compared to RealNet using only 500 anomaly samples as shown in submission Tab. 3.
>
> 2. Similarly, AnomalyAny, as a general anomaly generation approach, can generate unseen anomalies under text guidance. Note that the realism of AnomalyAny’s generated samples refers primarily to visual appearance, but they may differ from actual anomaly patterns, leading to an inevitable training–testing domain gap. Such methods require a large number of anomaly patterns to sufficiently model the anomaly distribution boundaries. Using the officially reported MVTec AD detection performance, as compared in **Response to Review FW9K.1**, UniAG outperforms AnomalyAny with a large margin.
>
> We emphasize that, for FSAG methods, generating samples that closely resemble the task-specific anomaly patterns is crucial for downstream anomaly detection performance, which unsupervised or zero-shot approaches cannot reliably guarantee.
>
> ## 3. More Visual Results on Composite Dataset.
> We updated the comparison on a combined dataset consisting of VisA’s **full category** set and two medical datasets (HeadCT and BrainMRI), along with corresponding generated-sample visualizations. For convenience, and without violating the anonymity policy, **we provide partial visualizations of generated samples on the combined dataset in [part1](https://imgshare.cc/zmj2bhq2) and [part2](https://imgshare.cc/2cwjegdd)** and highlight UniAG-generated samples using yellow, red, and green to indicate VisA, BrainMRI, and HeadCT, respectively. We further demonstrate the generative diversity achieved on the HeadCT dataset.
>
> The updated tables and visualization results will be included in the revised submission.

---

> ### Author Response · Authors · 2025-11-20
> **Response to Reviewer buaZ. (Part 2)**
>
> ## 4. Incremental Learning for New Anomaly Category
> Similar to the multi-class anomaly detection setting[5,6], UniAG aims to train a single generative model across multiple anomaly categories, enabling category- and location-controlled anomaly generation while significantly reducing training and deployment costs. As an FSAG method, we  discuss the advantages of UniAG over existing single-class anomaly generation approaches in *Supp.Sec.B*, namely:
> 1. Improved training efficiency. on MVTec AD, we train only **one** model instead of **73** one-for-one FSAG models, reducing training time by **95.4%** compared to DualAnoDiff (CVPR 2025)[9];
> 2. Controllable anomaly generation.  Deploying one model, UniAG supports  users to specify the anomaly category and location to generate **realistic** and **diverse** anomaly data, which is beneficial for targeted optimization in practical applications;
> 3. Theoretically supports anomaly texture transfer across different categories to improve diversity, *i.e.*, using similar anomaly patterns from other categories to generate anomaly data for the current category, see the experiments and discussion in **Response to Review V4Rd.1**.
> 4. Achieves superior performance compared to existing one-for-one anomaly generation methods, both in terms of anomaly generation quality and subsequent anomaly detection performance;
>
> In contrast, incremental learning aims to train a model over a sequence of tasks or data segments **without accessing the entire past data**, while **preserving previously acquired knowledge** and **effectively integrating new information**. To the best of our knowledge, there are currently no existing works that strictly follow the incremental learning setting for anomaly generation, although a few studies have explored incremental learning for anomaly detection [10], typically at the cost of reduced performance compared to full-data training but with improved practical applicability. We believe incremental anomaly generation is a paradigm between UniAG and Zero-Shot anomaly generation and the detection capability on the generated data tends to decrease while the generalization capability increases (FSAG-> Incremental AG -> Zero-Shot AG). As a promising direction, we consider it part of our future work.
>
> ## 5. Are the compared works on BrainMRI and HeadCT the SOTA methods in the medical field?
> Following existing methods, we primarily focus on industrial anomaly generation and detection. The additional composite datasets including two medical datasets provided in the Appendix are intended to validate UniAG’s cross-domain compatibility in multi-class settings. UniAG achieves superior anomaly generation and detection performance compared to existing single-class generation state-of-the-art FSAG methods[7-9].
>
>
> [1] Zhang et al. Prototypical Residual Networks for Anomaly Detection and Localization. CVPR2023.
>
> [2] Zavrtanik et al. Draem-a discriminatively trained reconstruction embedding for surface anomaly detection. CVPR2021.
>
> [3] Li et al. Cutpaste: Self-supervised learning for anomaly detection and localization. CVPR2021.
>
> [4] Zhang et al. Realnet: A feature selection network with realistic synthetic anomaly for anomaly detection. CVPR2024.
>
> [5] You et al. A unified model for multi-class anomaly detection. NeurIPS2022.
>
> [6] He et al. Mambaad: Exploring state space models for multi-class unsupervised anomaly detection. NeurIPS2024.
>
> [7] Hu et al. AnomalyDiffusion: Few-Shot Anomaly Image Generation with Diffusion Model. AAAI2024.
>
> [8] Guan et al. Few-Shot Anomaly-Driven Generation for Anomaly Classification and Segmentation. ECCV2024.
>
> [9] Jin et al. Dual-Interrelated Diffusion Model for Few-Shot Anomaly Image Generation. CVPR2025.
>
> [10] Xiao et al. Imputation-based time-series anomaly detection with conditional weight-incremental diffusion models. ACM SIGKDD 2023.

---

### Official Review · Reviewer_FW9K · 2025-11-01

**Soundness:** 2
**Presentation:** 3
**Contribution:** 2
**Rating:** 4
**Confidence:** 4

**Summary:**

This paper proposes UniAG, a unified model capable of generating realistic and diverse anomalies across multiple categories. Specifically, it introduces a deep copy-paste anomaly generation strategy, in which a Spatial-Texture Alignment Diffusion Model (STA-DM) learns to fill local region masks with anomaly textures corresponding to user-specified categories. The paper further proposes a novel conditioning mechanism with explicit spatial–category guidance, instead of relying on text embeddings, enabling more realistic and diverse anomaly synthesis. Experimental results on the MVTec AD dataset demonstrate the effectiveness of the method.

**Strengths:**

1. The method is designed to address several issues in prior work, and introduces a unified model.

2. The writing is relatively clear and fluent.

**Weaknesses:**

1. Table 2 results are not very competitive. Even under the 4-shot setting with both normal and anomalous samples, the performance is still noticeably below recent FSAD SOTA methods that use only 4 normal shots (e.g., AnomalyAny, PromptAD).

2. The dataset is limited, experiments are conducted only on MVTec AD.

3. The experimental pipeline is relatively complex and may have limited applicability. Training and synthesis rely on foreground masks; the paper uses foreground masks from Zhang et al. (2023a) to locate object foregrounds in normal images. In other domains, accurate masks may not be available.

4. The paper notes that DualAnoDiff is incompatible with FSN-style reconstruction detectors because it cannot provide a background-aligned normal counterpart; this should be flagged as a setup difference rather than evidence of inferiority.

5. Minor issue: there are several spelling errors in the paper (e.g., “performancew” in the caption of Figure 1).

**Questions:**

see weaknesses:

1. Table 2 results are not very competitive. Even under the 4-shot setting with both normal and anomalous samples, the performance is still noticeably below recent FSAD SOTA methods that use only 4 normal shots (e.g., AnomalyAny, PromptAD).

2. The dataset is limited, experiments are conducted only on MVTec AD.

3. The experimental pipeline is relatively complex and may have limited applicability. Training and synthesis rely on foreground masks; the paper uses foreground masks from Zhang et al. (2023a) to locate object foregrounds in normal images. In other domains, accurate masks may not be available.

4. The paper notes that DualAnoDiff is incompatible with FSN-style reconstruction detectors because it cannot provide a background-aligned normal counterpart; this should be flagged as a setup difference rather than evidence of inferiority.

5. Minor issue: there are several spelling errors in the paper (e.g., “performancew” in the caption of Figure 1).

**Details Of Ethics Concerns:**

No ethics concerns.

---

> ### Author Response · Authors · 2025-11-14
> **Response to Reviewer FW9K.**
>
> We sincerely appreciate your valuable feedback. We address your concerns as follows. If any part of our response remains unclear or incomplete, please feel free to point it out.
> ## 1. Tab. 2 results are not very competitive
> In Tab. 2,  we evaluate AD performance using only **100** generated samples (the same ones used in Tab. 1) to amplify the generated sample quality from different FSAG methods on detection performance metrics.
>
> **In Tab. 3**, we verify **better anomaly localization performance** (Pixel AUROC and AP) of UniAG against existing **state-of-the-art full-shot unsupervised anomaly detection methods**, including Patchcore, SimpleNet, and RealNet, using **500** generated anomaly samples.
>
> Results in Tab. 3 align with the advantages of supervised anomaly detection based on FSAG-generated data over unsupervised anomaly detection methods, as acknowledged by [1-5] and noted in lines 46–47 of our paper.  Furthermore, Fig. 6 (right) in paper compares  AD performance against DualAnoDiff under different amounts of generated samples. Within a certain range,  AD performance exhibits a **scale law** with generated data volume.
>
> Regarding the few-shot unsupervised methods you mentioned, we report the AD performance comparison on MVTec AD as follows:
>
> | Methods | Image-AUC |Pixel-AUC |
> |----------|----------|----------|
> | PromptAD[8] (4-shot normal)   | 96.6             | 96.5   |
> | AnomalyAny[9] (4-shot anomaly) | 96.4            | 96.2  |
> | UniAG-UNet (4-shot anomaly)    | 97.4             | 98.2|
> | UniAG-FSN (4-shot anomaly)    | **98.6**       | **99.2** |
>
> In fact, PromptAD[8]—as a few-shot normal-sample anomaly detection method—does not match the performance of the full-shot normal-sample baselines reported in Tab. 3 of our paper.
> AnomalyAny[9] focuses on generating unknown anomalies (no anomaly reference) and aims to produce visually plausible anomalous samples rather than samples consistent with real anomaly distributions. Therefore, even under the same 4-shot setting, UniAG achieves superior anomaly detection performance from generated anomaly samples that are more realistic, diverse, and distribution-consistent with the real anomaly references.
>
> ## 2. Limited experimental dataset on MVTec AD.
> As stated in lines 359–361 and 366-368, rigorously following current FSAG works [1–5], we conduct few-shot experiments primarily on comprehensive MVTec AD. To further demonstrate the generality of UniAG, we report results on a composite dataset (VisA+medical) in the initial submission *Supp.Sec.C*. Please refer to revised submission Tab.5, Tab.6, and visualized generated samples in Fig. 13 for more details.
>
> ## 3. Complex Pipeline and Foreground-mask Dependent
> 1). In fact, compared with unsupervised anomaly detection, FSAG methods are inherently more complex and require **paired anomaly–mask generation**, **sample verification/filtering**, and **detector training**. To ensure a fair comparison, UniAG closely follows the established FSAG pipeline [2-3],   superior to DualAnoDiff(CVPR2025) in terms of training and generation efficiency, as shown in Fig.1.
>
> 2). Importantly, **ALL** FSAG methods rely on spatial information to obtain **diverse and reasonable** anomaly masks.
>
> Following prior local anomaly-blending and inpainting approaches [2,7], we use a foreground prior to ensure that masks lie within the foreground region.  Other FSAG methods that generate masks [1,2,4] do not use the foreground mask directly, but instead require additional models to learn mask distributions and derive implicit spatial cues—substantially increasing training and generation overhead.
>
> We provide a detailed comparison of mask-generation strategies across existing FSAG methods, along with a discussion on their advantages and limitations. Please refer to  Supplementary *Sec. E* for details.
>
>
> ## 4. DualAnoDiff's incompatibility is more difference rather than inferiority.
> The statement “DualAnoDiff is incompatible with reconstruction-based FSN” appears in the caption of Tab. 3. We intend to clarify why DualAnoDiff + FSN is not included as a comparison entry, despite UniAG-generated samples (paired with a U-Net detector) achieving higher detection performance than DualAnoDiff. We agree with the reviewer that this incompatibility mainly stems from differences in methodological settings rather than an inherent limitation of DualAnoDiff.
>
> ## 5. Minor issue.
> Thank you for pointing out this spelling error; we will check and correct it in the revised version.

---

> > ### Author Response · Authors · 2025-11-14
> > **References**
> >
> > [1] Duan et al. Few-Shot Defect Image Generation via Defect-Aware Feature Manipulation. AAAI2023.
> >
> > [2] Hu et al. AnomalyDiffusion: Few-Shot Anomaly Image Generation with Diffusion Model. AAAI2024.
> >
> > [3] Guan et al. Few-Shot Anomaly-Driven Generation for Anomaly Classification and Segmentation. ECCV2024.
> >
> > [4] Jin et al. Dual-Interrelated Diffusion Model for Few-Shot Anomaly Image Generation. CVPR2025.
> >
> > [5] Xu et al. FAST: Foreground-aware Diffusion with Accelerated Sampling Trajectory for Segmentation-oriented Anomaly Synthesis. NeurIPS2025.
> >
> > [6] Dai et al. SeaS: Few-shot Industrial Anomaly Image Generation with Separation and Sharing Fine-tuning. ICCV2025.
> >
> > [7] Zhang et al. Prototypical Residual Networks for Anomaly Detection and Localization. CVPR2023.
> >
> > [8] Li et al. PromptAD: Learning Prompts with only Normal Samples for Few-Shot Anomaly Detection. CVPR2024.
> >
> > [9] Sun et al. Unseen Visual Anomaly Generation. ICCV2025.

---

### Official Review · Reviewer_V4Rd · 2025-11-01

**Soundness:** 2
**Presentation:** 2
**Contribution:** 2
**Rating:** 4
**Confidence:** 3

**Summary:**

This paper presents UniAG, a novel approach designed to address the issue of lacking anomaly samples in multi-class anomaly detection tasks. Traditional methods require training a separate model for each anomaly category, resulting in low training and deployment efficiency. Besides, GANs or copy-and-paste methods often generate anomalous samples that lack authenticity or diversity. UniAG is a diffusion-based few-shot anomaly generation framework that uses a learnable local “deep copy-paste” strategy plus explicit spatial–category conditioning to generate realistic, diverse multi-class anomalous patches from only a few examples, improving downstream detection, localization and classification.

Contributions:
1. Proposes UniAG, the first (practical) unified FSAG model that generates multiple anomaly categories with a single network (avoids one-model-per-class).
2. Introduces a deep copy–paste local generation pipeline: learnable anomalous patch synthesis + blending into normal backgrounds to preserve background fidelity and reliably synthesize small anomalies.

**Strengths:**

Strengths:
1. Unified Multi-Class Generation: Enables a single model to support up to 73 anomaly categories on MVTec-AD dataset, resolving the training and deployment burden of "one model per category," facilitating industrial applications and model maintenance.

2. Explicit Conditional Design: Spatial-Category Embedding. Replaces ambiguous textual conditions with explicit spatial + category conditions, providing the diffusion model with precise local location and category signals, thereby improving the generation and category consistency of small-region anomalies.

3. Anomaly Injection Adapter and Local Training Strategy: By injecting a trainable branch AIA and local crop/affine enhancements into UNet, the model can focus on learning local texture distributions, generating diverse and realistic anomaly patches, and stably blending them onto a normal background.

4. Scalability and Data Efficiency: The authors present experiments on scalability with increasing number of categories and data volume, showing that the model experiences minimal performance degradation and significant gains as the number of categories and generated samples increases, indicating that the method has good adaptability.

**Weaknesses:**

Weaknesses:

1. Still relies on a small number of real anomaly samples and their masks.

    I'm not saying that FSAG's task is unreasonable, but your method seems to heavily rely on anomalous images from a single category, failing to transfer anomalous data across categories. Especially for texture data, can scratches from wood be transferred to tiles? Can cracks from tiles be transferred to wood? I believe the core goal of FSAG isn't to generate a large number of anomalous images from a small number of anomalous images, because for newly designed, newly produced categories, we struggle to even obtain a single anomalous image. It seems that through cross-category transfer, we can generate some anomalous samples for model training.

2. Sensitive to mask/foreground positioning quality.

    SCE relies on accurate local masks; if the mask is biased (boundary misalignment, missing label), the generated texture may be misaligned or produce boundary artifacts, affecting downstream detection. I think what you absolutely must consider is the accuracy of the labels. Due to the nature of the FSAG task, only a very small number of anomalous images can be used. Therefore, if noisy labels appear in these anomalous images, the features learned by the model will be severely affected.
    Mitigation suggestions: Add mask pertubation augmentation, post-processing fusion (improved Poisson blending/seamless cloning), or add mask noise simulation during training.

3. Possible generation-detection bias.

    While UniAG prioritizes realism, synthetic samples may still exhibit micro-statistical differences from real anomalies, and the trained detector may be sensitive to these differences. In your qualitative experimental results, I can clearly see the lack of realism in the carpet class and some other categories. A clear boundary between the anomalous and normal regions may not be what we want. Cut-paste is the first step; how to make the final generated result smoother is a question that needs to be considered.

**Questions:**

1. Can scratches from wood be transferred to tiles? Can cracks from tiles be transferred to wood?
2. I want to see if, under the 4-shot task setting, the label of one image is not accurate enough, which would lead to a significant decline in the experimental results.
3. Can you resolve the issue of insufficient realism in the generated images, especially in the carpet category?

---

> ### Author Response · Authors · 2025-11-18
> **Response to Reviewer V4Rd. (Part1)**
>
> Thank you for your review and thoughtful feedback. We provide our responses to your concerns below.
> ## 1. Reliance on Limited Real Anomaly Samples and Cross-Category Anomaly Transfer
> Similar to [1–6], our goal is to generate realistic and diverse anomalies under extremely limited anomaly guidance. Considering the scarcity of anomalous data, our approach in fact uses **much fewer** anomaly prompts than existing FSAG methods—only 4-shot instead of 1/3 of all anomaly samples (**4*73=292** vs. **1258/3=419** prompts for 73 categories in total) as in [2–6]. This reduces reliance on anomaly cues and further demonstrates the practical applicability of FSAG under real-world constraints.
>
> We agree that cross-category anomaly transferring is a promising approach for zero-shot unseen anomaly generation [7,8]. However, the latest method, AnomalyAny[8] shows AD performance (image-AUROC, pixel-AUROC) $(1.0,2.0)\downarrow$ lower than UniAG in (**Response to Reviewer FW9K.1**). We argue that relying only on cross-category anomaly transferring has inherent limitations: **it heavily relies on contextual priors**, making it unclear which categories can serve as valid sources; it **struggles to handles domain-specific rare anomalies** with no meaningful cross-category equivalents; and without real anomaly guidance, generated samples, though visually plausible, **exhibit a domain gap with the real test anomalies**, offering limited training utility and sub-optimal AD performance.
>
> To address your question about transferring anomalies across categories (e.g., wood scratch → tile crack, tile crack → wood scratch), we conduct targeted experiments under the same settings (UniAG+U-Net, single anomaly), and evaluate AD performance shown below.
>
> | Anomaly Pattern Source(number) | Target Anomaly (detection) |Image-AUC | Pixel-AUC |
> |----------|----------|----------|----------|
> | wood_scratch（100） | wood_scratch  | **98.4**  | *96.5*|
> | tile_crack（100） | wood_scratch | 96.2  | 94.9|
> | wood_scratch（100）+ tile_crack（100）   | wood_scratch | **98.4** | **96.7**|
> | tile_crack（100）    | tile_crack       | **100.** | *98.3* |
> | wood_scratch（100）    | tile_crack       | 98.2 | 96.0|
> | tile_crack（100）+wood_scratch（100）| tile_crack | **100.** | **98.4** |
>
> Cross-category texture transfer produces reasonable anomaly detection results but still lags behind UniAG, which leverages 4-shot anomaly prompts to generate samples closer to real test anomalies. Nevertheless, such transfers can complement UniAG by adding diversity and potentially further improving anomaly localization when combined with its generated realistic samples.
>
> ## 2.Sensitive to imperfect mask quality.
> In FSAG, the mask defines the anomaly pattern and also drives subsequent (image, mask) generation. Thus, **not only SCE but all existing FSAG methods depend on accurate anomaly masks**. As with prior works[1-6], we follow the standard assumption that anomaly datasets provide reasonably aligned masks in MVTec AD dataset.
>
> We appreciate the reviewer’s concern regarding potential mask bias and, on the MVTec AD dataset, use the interior of the original mask’s bounding box as an inconsistency mask (b-box). Then, using the same experimental settings as paper Tab.2, we compared the anomaly detection performance against DualAnoDiff(CVPR2025)[4] on the first three alphabetically sorted categories (bottle, cable, capsule) of MVTEC AD. The AD performance (evaluated with gt masks) results are as follows:
>
> | Mask | Image-AUC |Pixel-AUC |
> |----------|----------|----------|
> | DualAnoDiff(b-box) | 91.1($\downarrow 1.9$) | 80.6($\downarrow 1.5$) |
> | DualAnoDiff(gt mask) |93.0  | 82.1|
> | UniAG(b-box) | 96.5 ($\downarrow 0.3$) | 86.9 ($\downarrow 0.1$)  |
> | UniAG(gt mask) | 96.8 | 87.0|
>
> UniAG shows only a minor performance drop (0.3, 0.1) under inconsistent masks, compared to DualAnoDiff (1.9, 1.5), demonstrating lower sensitivity. This is because UniAG uses spatial–category aligned embeddings to **explicitly constrain anomaly generation within the given mask, providing an upper bound on mask error**. In contrast, DualAnoDiff generates local anomalies globally and then conducts foreground segmentation, which may amplify mask misalignment due to **anomaly content confusion and the lack of an error upper bound**.

---

> ### Author Response · Authors · 2025-11-18
> **Response to Reviewer V4Rd.  (Part2)**
>
> ## 3. Possible Generation-Detection Bias
> 1. Realism and Diversity Trade-off. UniAG aims to generate anomaly samples that are both **realistic** and **diverse**, which are essential for detecting real anomalies [1-7]. Under few-shot references, realism and diversity involve a trade-off. UniAG addresses this by applying spatial and apparent augmentations (e.g., affine transforms, color jittering) to local patterns, enhancing diversity while preventing mode collapse. Some micro-statistical differences from real anomalies are unavoidable but help extend the few-shot distribution and improve generalization. Overall, our deep copy–paste strategy combined with spatial–category aligned embeddings achieves **superior realism (FID ↓ 1.30, IS ↑ 0.13) and diversity (IC-LPIPS ↑ 0.02)**, while **preserving foreground–background consistency** (Tab.1, Fig.5).
>
> 2. Possible Clear Boundary Issue. Unlike semantic recognition tasks, anomaly detection focuses on identifying **local texture inconsistencies**. In this context, **local gradient discontinuities also constitute a valid anomaly pattern** and naturally **align with the anomaly mask**. Importantly, the proportion of **single-pixel boundary introduced by simple paste operation is negligible** compared with the area of the **human-annotatable anomalous texture**, thus having **little effect** on subsequent supervised anomaly segmentation or reconstruction-based anomaly detection methods. This has already been confirmed in prior works that adopt the same "Simple Pasting" blending to construct anomalies[10–13]. In *Supp.Sec.E.3* and results with additional comparison in **Response to Reviewer buaZ.1**, we compare several strategies for removing clear boundaries and additionally include a new **Gaussian-Feathered Boundary Blending**. The results show that these techniques tend to **weaken real anomalous patterns**, causing **uncertain inconsistencies between the anomaly regions and the paired mask**, and ultimately leading to **suboptimal anomaly detection performance**.
> ## 4. Questions.
>
> 1. See comparison and analysis in 1.
>
> 2. See comparison and analysis in 2.
>
> 3. See comparison and analysis in 3.
> We are unsure which image you are referring to regarding the lack of realism in the generated carpet category (in Fig.2,3,5,7,11). If possible, please indicate the specific instance so that we can investigate it further.
>
> [1] Duan et al. Few-Shot Defect Image Generation via Defect-Aware Feature Manipulation. AAAI2023.
>
> [2] Hu et al. AnomalyDiffusion: Few-Shot Anomaly Image Generation with Diffusion Model. AAAI2024.
>
> [3] Guan et al. Few-Shot Anomaly-Driven Generation for Anomaly Classification and Segmentation. ECCV2024.
>
> [4] Jin et al. Dual-Interrelated Diffusion Model for Few-Shot Anomaly Image Generation. CVPR2025.
>
> [5] Xu et al. FAST: Foreground-aware Diffusion with Accelerated Sampling Trajectory for Segmentation-oriented Anomaly Synthesis. NeurIPS2025.
>
> [6] Dai et al. SeaS: Few-shot Industrial Anomaly Image Generation with Separation and Sharing Fine-tuning. ICCV2025.
>
> [7] Shi et al. Few-shot defect image generation based on consistency modeling. ECCV2024.
>
> [8] Sun et al. Unseen Visual Anomaly Generation. ICCV2025.
>
> [9] Zhou et al. AnomalyCLIP: Object-agnostic Prompt Learning for Zero-shot Anomaly Detection. ICLR2024.
>
> [10] Zhang et al. Prototypical Residual Networks for Anomaly Detection and Localization. CVPR2023.
>
> [11] Zavrtanik et al. Draem-a discriminatively trained reconstruction embedding for surface anomaly detection. CVPR2021.
>
> [12] Li et al. Cutpaste: Self-supervised learning for anomaly detection and localization. CVPR2021.
>
> [13] Zhang et al. Realnet: A feature selection network with realistic synthetic anomaly for anomaly detection. CVPR2024.

---

### Author Response · Authors · 2025-11-25
**Global Response and Revised Manuscript Highlights (Part 2)**

# Highlights of the Revised Manuscript

1. We add comparisons of anomaly generation and detection performance on the cross-domain composite dataset (industrial anomaly dataset ViSA, medical dataset of HeadCT and BrainMRI) in Tables 6 and 7 of the main text.
2. We supplement comparisons with SeaS[5] in Tables 1, 2, 3, 4, 6, 7, and Figure 5 of the main text.
3. In Table 8 and Figure 8 of the main text, we provide quantitative and qualitative comparisons of different fusion methods, accompanied by more detailed and accurate analytical discussions.
4. In response to the reviewers' suggestions, we present cross-domain anomaly transferring analysis and comparisons in Figure 10 of the supplementary material, performance comparison under masking inconsistency in Table 10, and generation visualization results on additional constituent datasets (including VisA, BrainMRI, and HeadCT) in Figure 13.
5. We have corrected some typos and ambiguous expressions.

[1] Duan et al. Few-Shot Defect Image Generation via Defect-Aware Feature Manipulation. AAAI2023.

[2] Hu et al. AnomalyDiffusion: Few-Shot Anomaly Image Generation with Diffusion Model. AAAI2024.

[3] Guan et al. Few-Shot Anomaly-Driven Generation for Anomaly Classification and Segmentation. ECCV2024.

[4] Jin et al. Dual-Interrelated Diffusion Model for Few-Shot Anomaly Image Generation. CVPR2025.

[5] Dai et al. SeaS: Few-shot Industrial Anomaly Image Generation with Separation and Sharing Fine-tuning. ICCV2025.

[6] Sun et al. Unseen Visual Anomaly Generation. ICCV2025.

[7] Zhang et al. Prototypical Residual Networks for Anomaly Detection and Localization. CVPR2023.

[8] Zavrtanik et al. Draem-a discriminatively trained reconstruction embedding for surface anomaly detection. CVPR2021.

[9] Li et al. Cutpaste: Self-supervised learning for anomaly detection and localization. CVPR2021.

[10] Zhang et al. Realnet: A feature selection network with realistic synthetic anomaly for anomaly detection. CVPR2024.

---

### Author Response · Authors · 2025-11-25
**Global Response and Revised Manuscript Highlights (Part 1)**

First, we would like to express our sincere gratitude to all the reviewers for their dedicated efforts and valuable comments and suggestions. Herein, we first provide a brief response to several common concerns (for detailed information, please refer to the full Response, respectively), followed by the highlights of the revised manuscript incorporating the reviewers' feedback.
# Common Issue Response:
1. **Reliance on Limited Real Anomaly Samples.** UniAG is designed to generate realistic, diverse anomaly samples with few-shot anomalous references, boosting supervised anomaly detection performance—consistent with existing FSAG methods [1-4]. As clearly established in prior work, FSAG methods offer inherent advantages over Unsupervised Anomaly Detection (UAD), including **superior anomaly localization** and **support for fine-grained anomaly classification**, with better practical deployability. Our results in Tab. 3 demonstrate that UniAG, combined with a simple U-Net, outperforms state-of-the-art UAD methods in localization. Tab. 4 further presents anomaly classification performance using generated data— a capability unavailable in UAD. Compared to reference-free generation methods based on texture transfer or generative fine-tuning, **FSAG not only ensures visually realistic anomalies but also produces training samples aligned with the test distribution, reducing the train-test domain gap** for enhanced detection performance.
2. **Experiments are conducted only on MVTec AD.** Following the experimental and comparison settings of [1-4], existing methods only conduct comprehensive task comparisons on MVTec AD (encompassing 73 anomaly categories). Due to space constraints, we initially included the performance of UniAG on a **cross-domain dataset—comprising the industrial dataset ViSA and two medical datasets (BrainMRI, HeadCT)**—in the supplementary material to verify its generalization ability. In the revised version, this content has been moved to the main text.
3. **Dependency on Foreground Masks.** We adopt the approximate foreground region used in previous work [7] to filter out the augmented mask regions outside the target, thereby ensuring that anomalies are confined within the object. Noteworthy, **with object spatial misalignment, additional object location information is necessary**. For instance, [3] also employs a similar preset foreground filtering mask, while [2,5] rely on an extra mask generation module to obtain implicit spatial information, and [4] utilizes an additional segmentation module to derive the corresponding mask from the generated anomaly map. In contrast, the use of object-approximate foreground regions does not compromise fairness and simplifies the generation process.
4. **Simple Pasting Blending Issue.** In the initial submission, we compared 3 seamless blending approaches and found that **the simplest method yielded the most effective results**—direct copying achieved the best anomaly detection performance. During the rebuttal phase, we further supplemented the experiment with Gaussian feathering blending (a technique that only blends edge regions to eliminate obvious boundaries), yet our conclusion remained unchanged. This is because anomaly detection inherently focuses on **local texture inconsistencies: even local gradient discontinuities naturally constitute valid anomalies that spatially-aligned with the mask**. The single-pixel boundaries introduced by direct copying are negligible relative to human-annotatable anomalous regions, as confirmed by prior works[7-10].
# Re-emphasizing Our  Contributions
1. We analyze existing FSAG methods' limitations and propose a simple yet effective "deep copy-paste" strategy, which aligns normal backgrounds and local anomalous foregrounds with real domain distributions to boost anomaly detection performance.
2. We propose the Spatial-Texture Alignment Diffusion Model (STA-DM), which uses a novel **spatial-category embedding** for anomaly generation. Compared with conventional textual conditions, this embedding has **two key advantages**:
(1) Guided by explicit category ids and spatial regions, the model generates precise target textures within specified masks. Ignoring texture-irrelevant content, it greatly enhances the authenticity of generated anomalies. (2) **A unified model enables category- and location-controllable generation of all anomalies**, reducing training/deployment overhead and boosting flexibility effectively.
3.  Extensive experiments on MVTec AD and cross-domain dataset (VisA, BrainMRI, HeadCT) show UniAG outperforms peers in anomaly generation and downstream tasks (detection, localization, classification). On MVTec AD, it hits new SOTA in anomaly localization (AUROC/AP: 99.2/81.0) with only 4 anomaly examples and 500 generated samples per anomaly. Compared to the latest FSAG method DualAnoDiff[4], UniAG cuts training time to **1/20**, speeds up generation by **20×**, and has minimal inference GPU memory overhead.

---

### Author Response · Authors · 2025-11-28
**Request for Follow-Up Comments During Rebuttal Phase**

Dear Reviewers,

Thank you again for your time and efforts in evaluating our manuscript. As we are currently in the rebuttal phase, we would kindly appreciate it if you could provide your further comments or clarifications at your earliest convenience. This will help us continue the discussion and ensure that we address all concerns thoroughly within the given timeline.

Thank you very much for your support and consideration.

Best regards

---

### Note · Authors · 2026-01-26

I have read and agree with the venue's withdrawal policy on behalf of myself and my co-authors.

---

### Meta-Review · Area_Chair_mqPY · 2026-01-08

**Summary:**

Multiple reviewers flagged limited novelty, arguing UniAG largely integrates existing diffusion/inpainting + conditioning components. Others emphasized dependence on accurate masks/foreground priors, raising robustness and deployment concerns. Last, Comparisons to recent 2025 SOTA and broader multi-shot/generalization studies were requested.

**Reviewer Concerns:**

**Addressed:**
* The rebuttal strengthened the paper by empirically validating cross-category anomaly transfer.
* Mask noise sensitivity was directly tested using inconsistent/bounding-box masks, with only minor performance degradation.
* Baseline and competitiveness claims were strengthened with new comparisons.


**Outstanding:**
* Novelty concerns remain debatable due to the impression of "strong engineering + good results."
* Motivation of FSAG remains fundamentally unresolved.
* Generalization concerns regarding non-MVTec results and broader shot settings.

**Reviewer Scores:**

The rejection rating by mHJL may be unlikely to change after the rebuttal. While the rebuttal answered technical questions and added comparisons to newer FSAG methods, the reviewer’s core concerns, including motivation of FSAG itself, lack of novelty, and narrow evaluation are largely unsolved. Concerns of V4Rd and FW9K are partially addressed by the additional experiments, while other issues, such as deployment, remain. buaZ felt positive about this paper, and the reviewers' concerns were addressed in rebuttal. The rating perhaps remains stable.

---

### Decision · Program_Chairs · 2026-01-26

Reject